# DT$^2$: Decision-Targeted Digital Twins

**Harry Amad** [1]   **Mihaela van der Schaar** [1]

## Abstract

A digital twin (DT) is a virtual model of a real-world system that can assist decision-making by simulating scenarios induced by different policies. However, typical machine learning-based DTs do not optimise for this use case. We prove that, when model capacity is limited, training DTs to minimise one-step transition errors can produce suboptimal models for ranking sets of policies according to a reward function. We further show that this holds empirically, even with expressive model classes. To address this, we introduce DT$^2$, a decision-targeted DT training paradigm. Firstly, DT$^2$ uses fitted Q-evaluation to estimate values of candidate policies from offline data. A DT is then trained to generate rollouts that preserve pairwise policy rankings derived from these proxy ground-truth values with an architecture-agnostic loss function. We empirically demonstrate the efficacy of our method across a range of settings and architectures. DT$^2$ consistently improves policy ranking and reduces decision regret during policy selection relative to conventional DT training, both for policies used during training and for unseen policies, while maintaining a good level of raw simulation fidelity.

## 1. Introduction

A digital twin (DT) is a virtual model of a real-world system that can simulate its dynamics. DTs are appealing in many domains, such as finance (Slepneva et al., 2021), climate science (Voosen, 2020), manufacturing (Rosen et al., 2015), energy (Ismail et al., 2024), agriculture (Purcell & Neubauer, 2023), robotics (Mazumder et al., 2023), and medicine (Katsoulakis et al., 2024), due to their ability to guide decision-making processes. This typically manifests

with a user deliberating over DT-generated simulations of potential policies (Tao et al., 2018; Corral-Acero et al., 2020) to select the best one, or establish a preference ordering. Much of the DT literature highlights this critical use case, asserting that a useful DT is one that *"aids decision making"* (Wagg et al., 2020), and *"[informs] decisions that realize value"* (National Academy of Engineering and National Academies of Sciences, Engineering, and Medicine, 2024). To enable such decision support, DTs must generate realistic simulations that allow for user interrogation and verification, while ensuring the learned dynamics lead to correct rankings of relevant policies.

In machine learning (ML)-based DTs, however, this goal of decision support is often overlooked. Typically, ML-based DTs involve a model that is trained to minimise a measure of one-step transition error, such as mean squared error (MSE) or negative log likelihood (NLL), across all variables and time points in a dataset of observed state-action trajectories (San et al., 2023; Kuang et al., 2024; Holt et al., 2024; Canzaniello et al., 2024; Makarov et al., 2025; Amad et al., 2025). While this can lead to high fidelity simulators, such direction-, variable-, and timepoint-agnostic objectives do not explicitly focus on correctly ranking policies, and they can misalign DTs with their primary downstream purpose by under-emphasising variables or temporal slices that are especially crucial for discriminating between policies.

For example, consider a medical DT, used to help clinicians treat septic shock. Despite large data availabilities in ICU settings, clinicians typically focus on only a few key indicators to make timely decisions (Pickering et al., 2013), like blood pressure and lactate levels for sepsis. A DT trained with a typical, variable-agnostic, simulation loss considers non-critical dimensions as equivalent to such key indicator variables. By wasting model capacity on largely irrelevant features, such a DT could provide suboptimal decision support, jeopardising patient health. For a visual example of how simulation training can lead to poor decisions due to being direction-agnostic, see Figure 1.

In this work, we emphasise that, for human-in-the-loop decision support, a DT need not be a perfect replica of previously observed reality, and we contend that DT construction should move away from such context-agnostic optimisation, towards what is best for decision support.

---

[1]Department of Applied Mathematics and Theoretical Physics, University of Cambridge, Cambridge, United Kingdom. Correspondence to: Harry Amad <hmka3@cam.ac.uk>.

*Proceedings of the 43$^{rd}$ International Conference on Machine Learning*, Seoul, South Korea. PMLR 306, 2026. Copyright 2026 by the author(s).

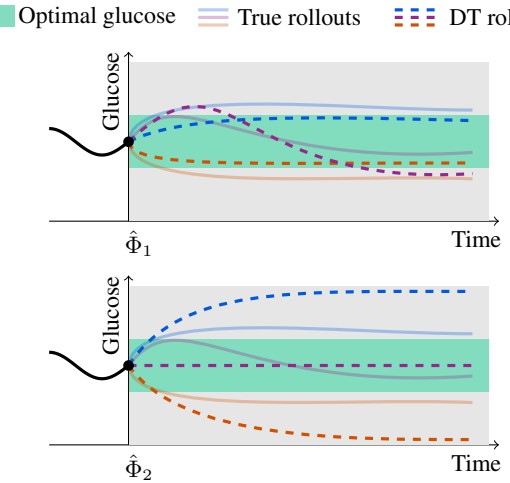

*Figure 1.* We visualise two diabetes DTs, $\hat{\Phi}_1$ (top) and $\hat{\Phi}_2$ (bottom), under treatment plans with **high**, **low**, and **adaptive** insulin levels, and we consider the 'value' of a treatment plan as the time it spends in the optimal glucose region. $\hat{\Phi}_1$ has lower MSE from the true rollouts, yet it leads to an incorrect treatment plan preference ordering (**high** $\approx$ **low** $\succ$ **adaptive**). In contrast, $\hat{\Phi}_2$ leads to a correct preference ordering (**adaptive** $\succ$ **high** $\approx$ **low**), despite its higher MSE, as its rollouts traverse the correct glucose regions.

For this purpose, we introduce DT$^2$: a **d**ecision-**t**argeted **DT** training paradigm that balances between seeking fidelity in general, and in those parts of the transition distribution that are most relevant for ranking candidate policies according to a known reward function. Since ground-truth rankings amongst candidate policies are not generally known *a priori*, we firstly estimate policy values using off-policy evaluation (OPE) methods, which can estimate scalar policy values from offline data (Fu et al., 2021) but, distinct from DTs, offer little in the way of interpretability, limiting their application in high-stakes domains (Rudin, 2019). We use these OPE estimates to construct proxy ground-truth pairwise rankings amongst candidate policies, and introduce a differentiable ranking loss function that operates alongside standard simulation objectives to encourage DTs to generate simulations that preserve these proxy rankings. By doing so, we distil the ranking ability of OPE into the interpretable structure of a DT. A tunable hyperparameter, $\lambda$, modulates the trade-off between raw simulation accuracy and reconstruction of OPE policy rankings. See code here https://github.com/harrya32/decision-targeted-dt and here https://github.com/vanderschaarlab/decision-targeted-dt. In summary, our contributions are:

1. We prove that standard DT training can be theoretically suboptimal for decision-making when the DT hypothesis space is limited, establishing motivation for decision-targeted training (§3).

2. We propose DT$^2$, an architecture-agnostic decision-targeted DT training paradigm that aligns simulations with proxy policy rankings (§4). We use a smooth approximation of Kendall's rank correlation coefficient (Kendall, 1938) (§4.1) to enable supervision with pairwise preference targets derived from OPE policy value estimates (§4.2).

3. We empirically demonstrate the alignment gap predicted by our theory (§6.1), and show that this persists even with highly expressive hypothesis spaces (§6.2). Across a suite of continuous control tasks, DT$^2$ reduces decision regret by more than 50% at an MSE cost of only 17%, demonstrating capacity for better decision support. Furthermore, we show that this advantage extends to out-of-distribution settings, achieving better preference orderings amongst policies never seen during training (§6.3).

## 2. Formalism

We now formalise the problem of decision support that we aim to address with DTs.

### 2.1. The Decision-Making Goal

Let a real-world system be modelled as a Markov Decision Process (MDP) denoted by $\mathcal{M} := (\mathcal{X}, \mathcal{A}, \Phi, r, \gamma)$. Here, $\mathcal{X} \subseteq \mathbb{R}^{d_{\mathcal{X}}}$ is a $d_{\mathcal{X}}$-dimensional state space, $\mathcal{A} \subseteq \mathbb{R}^{d_{\mathcal{A}}}$ is a $d_{\mathcal{A}}$-dimensional action space, $\Phi : \mathcal{X} \times \mathcal{A} \to \mathcal{P}(\mathcal{X})$ is the transition distribution, where $\mathcal{P}(\mathcal{X})$ denotes the set of probability measures over $\mathcal{X}$, $r : \mathcal{X} \times \mathcal{A} \times \mathcal{X} \to \mathbb{R}$ is a bounded reward function, and $\gamma \in [0, 1)$ is a discount factor. At time $t$, given the current state $\mathbf{x}_t \in \mathcal{X}$ and action $\mathbf{a}_t \in \mathcal{A}$, the next state is sampled according to $\mathbf{x}_{t+1} \sim \Phi(\cdot|\mathbf{x}_t, \mathbf{a}_t)$.

A policy $\pi : \mathcal{X} \to \mathcal{P}(\mathcal{A})$ can interact with $\mathcal{M}$ in an initial state $\mathbf{x}_0$, yielding a trajectory $\tau = (\mathbf{x}_0, \mathbf{a}_0, \mathbf{x}_1, \mathbf{a}_1, \dots)$, where $\mathbf{a}_t \sim \pi(\mathbf{x}_t)$, and we denote such $\tau \sim (\Phi, \pi)$. We define the value of a policy $\pi$ in $\mathcal{M}$ as the expected sum of discounted rewards:

$$J(\pi) := \mathbb{E}_{\tau \sim (\Phi, \pi)} \left[ \sum_{t=0}^{\infty} \gamma^t r(\mathbf{x}_t, \mathbf{a}_t, \mathbf{x}_{t+1}) \right]. \quad (1)$$

In decision-making processes, a user may want to interrogate trajectories induced by each $\pi$ in some candidate policy set $\Pi$, as well as establish a preference ordering over $\Pi$. This is defined by the relation $\succ$, where for any pair $\pi_i, \pi_j \in \Pi, \pi_i \succ \pi_j \iff J(\pi_i) > J(\pi_j)$. Alternatively, a reduced goal may be to identify the optimal policy $\pi^*$:

$$\pi^* := \arg\max_{\pi \in \Pi} J(\pi). \quad (2)$$

Generating such trajectories and determining such preference orderings with repeated experimentation on the real-world system is generally infeasible, especially in high-stakes domains such as medicine and public policy.

## 2.2. Digital Twins as Practical Surrogates

A DT can serve as a practical surrogate for the real system, involving a parametrised model $\hat{\Phi}_\theta$ that is designed to approximate $\Phi$. Given a policy $\pi$, the DT can simulate a trajectory from $\mathbf{x}_0$, denoted $\hat{\tau} = (\hat{\mathbf{x}}_0, \mathbf{a}_0, \hat{\mathbf{x}}_1, \mathbf{a}_1, ...) \sim (\hat{\Phi}_\theta, \pi)$. We denote the DT-estimated value of $\pi$ as:

$$\hat{J}(\pi; \theta) := \mathbb{E}_{\hat{\tau} \sim (\hat{\Phi}_\theta, \pi)} \left[ \sum_{t=0}^{\infty} \gamma^t r(\hat{\mathbf{x}}_t, \mathbf{a}_t, \hat{\mathbf{x}}_{t+1}) \right], \quad (3)$$

which leads to a DT preference ordering defined by $\hat{\succ}$, where $\pi_i \hat{\succ} \pi_j \iff \hat{J}(\pi_i; \theta) > \hat{J}(\pi_j; \theta)$, and a DT-optimal policy $\hat{\pi}^*$:

$$\hat{\pi}^* := \arg\max_{\pi \in \Pi} \hat{J}(\pi; \theta). \quad (4)$$

# 3. On the Misalignment of Conventional Digital Twin Training

Conventional ML-based DT training paradigms involve a training dataset $\mathcal{D} = \{(\mathbf{x}, \mathbf{a}, \mathbf{x}')_i\}_{i=1}^n$ of $n$ observed transitions collected from the real-world system, or related systems. The optimal DT parameters, $\theta^*$, are typically found by minimising a measure of simulation error, $\mathcal{L}_{\text{sim}}(\theta)$, over $\mathcal{D}$. There are a few possible instantiations of $\mathcal{L}_{\text{sim}}(\theta)$, with the most notable being NLL for probabilistic models:

$$\mathcal{L}_{\text{sim}}^{\text{NLL}}(\theta) := \mathbb{E}_{(\mathbf{x}, \mathbf{a}, \mathbf{x}') \sim \mathcal{D}} \left[ - \log \hat{\Phi}_\theta(\mathbf{x}' \mid \mathbf{x}, \mathbf{a}) \right], \quad (5)$$

or MSE when focusing on the expected trajectory or assuming constant variance:

$$\mathcal{L}_{\text{sim}}^{\text{MSE}}(\theta) := \mathbb{E}_{(\mathbf{x}, \mathbf{a}, \mathbf{x}') \sim \mathcal{D}} \left[ \|\mathbf{x}' - \hat{\mu}_\theta(\mathbf{x}, \mathbf{a})\|^2 \right], \quad (6)$$

where $\hat{\mu}_\theta(\mathbf{x}, \mathbf{a}) = \mathbb{E}_{\mathbf{x}' \sim \hat{\Phi}_\theta(\cdot | \mathbf{x}, \mathbf{a})}[\mathbf{x}']$. Such loss functions are a poor proxy for the ultimate goal of establishing preference orderings over policies.

**Theorem 3.1.** *Let $\Phi$ be the transition distribution of a real-world system, $\mathcal{D} = \{(\mathbf{x}, \mathbf{a}, \mathbf{x}')_i\}_{i=1}^n$ be a training dataset collected from $\Phi$, and $\mathcal{F} = \{\hat{\Phi}_\theta \mid \theta \in \Theta\}$ be a DT hypothesis class such that $\Phi \notin \mathcal{F}$ and*

$$\min_{\theta \in \Theta} \mathbb{E}_{(\mathbf{x}, \mathbf{a}) \sim \mathcal{D}} \left[ D_{\text{KL}}(\Phi(\cdot | \mathbf{x}, \mathbf{a}) \| \hat{\Phi}_\theta(\cdot | \mathbf{x}, \mathbf{a})) \right] > 0. \quad (7)$$

*Let $\hat{\Phi}_{\theta^*}$ be the $\mathcal{L}_{\text{sim}}$-optimal DT, i.e.*

$$\theta^* \in \arg\min_{\theta \in \Theta} \mathcal{L}_{\text{sim}}(\theta). \quad (8)$$

*Then, there exists a reward function $r$ and a set of policies $\Pi$ such that $\hat{\Phi}_{\theta^*}$ induces a suboptimal preference ordering on $\Pi$ according to $r$.*

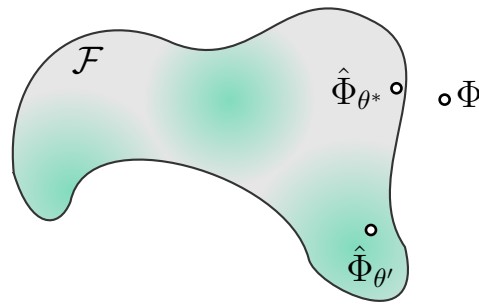

*Figure 2.* When $\Phi \notin \mathcal{F}$, the $\mathcal{L}_{\text{sim}}$-optimal DT, $\hat{\Phi}_{\theta^*}$, may land in a region of low decision quality for some candidate policy set $\Pi$ and reward function $r$ (grey areas). With decision-targeted DT training, we wish to find $\hat{\Phi}_{\theta'}$ that resides in a better decision region (green areas), at a small simulation fidelity cost.

*Proof.* See Appendix A.1. □

From Theorem 3.1 we can see that, when a DT cannot exactly model its target system, optimising $\mathcal{L}_{\text{sim}}$ will not generally prioritise decision-making. Given that collecting large data for certain real-world systems can be difficult, and that DTs should constantly update alongside their counterpart system (Amad et al., 2025), using lightweight model classes, such as equation-based DTs, or small neural DTs, is common, and here Theorem 3.1 can directly apply. Furthermore, even when expressive model classes are used, the complex dynamics or partial observability of real-world systems may still result in non-covering hypothesis spaces. Moreover, under a further assumption on $\mathcal{F}$, we can see that better models than the $\mathcal{L}_{\text{sim}}$-optimal DT exist for decision-making.

**Definition 3.2.** Let $(\bar{\mathbf{x}}, \bar{\mathbf{a}}) \in \text{supp}(\mathcal{D})$ and $G \subseteq \mathcal{X}$ be such that $\Phi(G \mid \bar{\mathbf{x}}, \bar{\mathbf{a}}) \neq \hat{\Phi}_{\theta^*}(G \mid \bar{\mathbf{x}}, \bar{\mathbf{a}})$. Define

$$p := \Phi(G \mid \bar{\mathbf{x}}, \bar{\mathbf{a}}), \; \hat{p} := \hat{\Phi}_{\theta^*}(G \mid \bar{\mathbf{x}}, \bar{\mathbf{a}}). \quad (9)$$

Assume, w.l.o.g., that $p > \hat{p}$. Then, $\mathcal{F}$ allows *local improvement at $\theta^*$* if there exists $\theta' \in \Theta$ such that

$$\hat{p}' := \hat{\Phi}_{\theta'}(G \mid \bar{\mathbf{x}}, \bar{\mathbf{a}}) > \hat{p}. \quad (10)$$

This definition applies to $\mathcal{F}$ when it contains an alternative to the $\mathcal{L}_{\text{sim}}$-optimal model whose predictions on a local subset of transitions are closer to the true transition distribution.

**Theorem 3.3.** *Assume the conditions of Theorem 3.1 and that $\mathcal{F}$ allows local improvement at $\theta^*$. Then there exists a reward function $r$, a set of policies $\Pi$, and $\theta' \in \Theta \setminus \{\theta^*\}$ such that $\hat{\Phi}_{\theta'}$ induces a strictly better preference ordering on $\Pi$ according to $r$ than $\hat{\Phi}_{\theta^*}$.*

*Proof.* See Appendix A.2. □

Theorem 3.3 outlines our theoretical motivation to search for a better training paradigm for DTs than optimising $\mathcal{L}_{\text{sim}}$,

that will better converge to a model that induces a good preference ordering on $\Pi$ according to $r$ (Figure 2). We elaborate on the scenarios when these theorems directly apply, and empirically validate them, in §6.1. Moreover, we hypothesise that, even when $\Phi \in \mathcal{F}$, optimising $\mathcal{L}_{\text{sim}}$ will poorly converge to an optimal decision-making model, and we show empirical evidence for this in §6.2 and §6.3.

## 4. DT$^2$: Decision-Targeted Digital Twin Training

To train a decision-targeted DT we must address some practical hurdles. First, comparing a ground-truth preference ordering $\succ$ to a DT-estimated $\hat{\succ}$ with standard ranking metrics involves indicator functions, e.g. checking $\mathbb{I}(\pi_i \succ \pi_j) = \mathbb{I}(\pi_i \hat{\succ} \pi_j) \; \forall \pi_i, \pi_j \in \Pi$, which are difficult to optimise, because they have zero gradient almost everywhere. We address this by using a smoothed ranking loss for decision-targeted training (§4.1). Second, the ground-truth $\succ$ is almost never available, as this is, in part, the reason for constructing a DT in the first place (§2.2). So, we derive proxy preference orderings using OPE policy value estimates to target during training (§4.2). Finally, to avoid the need to generate and backpropagate through long DT rollouts when optimising $\hat{\succ}$ during training, we employ a value-bootstrapped scheme (§4.3). This forms DT$^2$, our method for DT training that focuses on modelling dynamics that are critical for decision support.

### 4.1. Smoothed Ranking Loss

We define a ranking objective based on a smooth approximation of Kendall's rank correlation coefficient (Kendall, 1938). Let $\Delta_{ij} := J(\pi_i) - J(\pi_j)$ be the ground-truth value difference between two policies, and $\hat{\Delta}_{ij}(\theta) := \hat{J}(\pi_i; \theta) - \hat{J}(\pi_j; \theta)$ be the corresponding DT-estimated value difference. Kendall's correlation measures the concordance of signs between these differences:

$$\frac{1}{|\mathcal{C}|} \sum_{(i,j) \in \mathcal{C}} \text{sign}\left(\Delta_{ij}\right) \cdot \text{sign}\left(\hat{\Delta}_{ij}(\theta)\right), \quad (11)$$

where $\mathcal{C}$ is the set of pairs in a candidate policy set $\Pi$. To make this suitable for gradient-based optimisation, we replace the sign functions with hyperbolic tangents, and set the complement as our ranking loss to be minimised:

$$\mathcal{L}_{\text{rank}}(\theta) := 1 - \frac{1}{|\mathcal{C}|} \sum_{(i,j) \in \mathcal{C}} \tanh\left(\frac{\Delta_{ij}}{\alpha}\right) \cdot \tanh\left(\frac{\hat{\Delta}_{ij}(\theta)}{\alpha}\right) \quad (12)$$

where the temperature $\alpha$ governs the trade-off between faithfulness to the discrete Kendall correlation and gradient smoothness. Smaller $\alpha$ produces sharper, more sign-like curves, while larger $\alpha$ yields smoother gradients that can lead to more stable optimisation.

A key benefit of this formulation is that the $\tanh(\Delta_{ij}/\alpha)$ term acts as an implicit weighting mechanism. When the reference evaluator is indifferent between two policies ($\Delta_{ij} \approx 0$), the term $\tanh(\Delta_{ij}/\alpha)$ approaches zero, causing the gradient contribution for that pair to vanish. Consequently, the model is encouraged to ignore noise in policy separation, allocating capacity toward distinguishing between high-regret pairs. We empirically compare this formulation to alternative ranking losses in Appendix E.

### 4.2. Off-Policy Evaluation for Proxy Preference Targets

Since ground-truth policy values $J(\pi)$ are generally inaccessible in the offline setting we operate in, we derive proxy $\Delta_{ij}$ targets using principles from OPE (§5.4). Specifically, we use fitted Q-evaluation (FQE) (Le et al., 2019) to estimate the action-value function $Q^\pi(\mathbf{x}, \mathbf{a})$ for each $\pi \in \Pi$. $Q^\pi(\mathbf{x}, \mathbf{a})$ denotes the expected long-term return of executing action $\mathbf{a}$ in state $\mathbf{x}$ and following $\pi$ thereafter:

$$Q^\pi(\mathbf{x}, \mathbf{a}) := \mathbb{E}_{\tau \sim (\Phi, \pi)} \left[ \sum_{t=0}^{\infty} \gamma^t r_t \Big| \mathbf{x}_0 = \mathbf{x}, \mathbf{a}_0 = \mathbf{a} \right]. \quad (13)$$

FQE approximates this with a parameterised function $Q_\psi^\pi$ trained on $\mathcal{D}$ to minimise the expected Bellman error:

$$\mathcal{L}_{\text{FQE}}(\psi) := \mathbb{E}_{\mathcal{D}} \left[ \left( Q_\psi^\pi(\mathbf{x}, \mathbf{a}) - \left( r + \gamma Q_{\bar{\psi}}^\pi(\mathbf{x}', \pi(\mathbf{x}')) \right) \right)^2 \right] \quad (14)$$

where $Q_{\bar{\psi}}^\pi$ is a target network. The parameters $\bar{\psi}$ are frozen copies of $\psi$ that are updated periodically to stabilise the bootstrapping target. Once converged, the scalar value of $\pi$ can be estimated by averaging $Q_\psi^\pi$ over the initial states,

$$J_{\text{FQE}}(\pi) \approx \mathbb{E}_{\mathbf{x}_0 \sim \mathcal{D}, \, \mathbf{a} \sim \pi(\mathbf{x}_0)}[Q_\psi^\pi(\mathbf{x}_0, \mathbf{a})]. \quad (15)$$

These estimates allow us to construct the proxy pairwise rankings targets $\Delta_{ij}^{\text{FQE}} = J_{\text{FQE}}(\pi_i) - J_{\text{FQE}}(\pi_j)$ that are used in place of $\Delta_{ij}$ in Eq. 12.

We select FQE amongst the plethora of OPE methods given its simplicity and strong performance on OPE benchmarks (Fu et al., 2021; Voloshin et al., 2021). Furthermore, while FQE can be prone to systematic bias, particularly for out-of-distribution policies, these errors will tend to cancel out during pairwise comparisons. This makes FQE-based rankings potentially more robust than its scalar value estimates, as relative orderings can be preserved even when absolute value magnitudes are biased.

### 4.3. Bootstrapping Digital Twin Value Estimates

Computing the DT-estimated $\hat{\Delta}_{ij}(\theta)$ for Eq. 12 requires simulating trajectories $\hat{\tau} \sim (\hat{\Phi}_\theta, \pi)$ (Eq. 3). However, backpropagating through long rollouts to optimise $\mathcal{L}_{\text{rank}}$ is computationally expensive and prone to vanishing or exploding gradients (Bengio et al., 1994; Pascanu et al., 2013).

To mitigate this, we employ a bootstrapping scheme that truncates DT rollouts at a fixed horizon $H$ and approximates the remaining return by re-using a frozen $Q_\psi^\pi$ trained with FQE. The bootstrapped DT-value estimate for policy $\pi$ is defined as:

$$\hat{J}_H(\pi; \theta) := \mathbb{E}_{\hat{\tau} \sim (\hat{\Phi}_\theta, \pi)} \left[ \sum_{t=0}^{H-1} \gamma^t r_t + \gamma^H Q_\psi^\pi(\hat{x}_H, a_H) \right]. \tag{16}$$

By using $\hat{J}_H$ to compute $\hat{\Delta}_{ij}(\theta)$, we reduce the cost of calculating $\mathcal{L}_{\text{rank}}$, and improve its gradient stability.

To ensure $DT^2$ still permits generation of high fidelity trajectories, which is important for the interpretability aspects of human-in-the-loop decision support, the ultimate training objective combines $\mathcal{L}_{\text{rank}}$ with $\mathcal{L}_{\text{sim}}$:

$$\mathcal{L}_{\text{DT}^2}(\theta) := (1 - \lambda)\mathcal{L}_{\text{sim}}(\theta) + \lambda\mathcal{L}_{\text{rank}}(\theta). \tag{17}$$

The hyperparameter $\lambda \in [0, 1]$ modulates the trade-off between optimising for raw simulation fidelity and decision-targeting. $\lambda \to 0$ recovers standard DT training, while $\lambda \to 1$ yields a purely decision-targeted model. We investigate its effects in §6.4. By shaping the DT loss landscape with $\Delta_{ij}^{\text{FQE}}$, $DT^2$ distils the decision-making power of 'black-box' OPE methods into the interpretable structure of a DT.

# 5. Related Works

## 5.1. Digital Twins

1960s aerospace engineering pioneered the use of DTs (Allen, 2021), where physical laws can fully describe system dynamics, but manually-defining such DTs requires significant domain expertise. Methods like SINDy (Brunton et al., 2016) and PySR (Cranmer, 2023) aim to automate the discovery of such mechanistic models, but they are often outperformed by more expressive deep learning approaches.

Deep learning DTs approximate system dynamics from longitudinal data using a variety of model architectures, including standard MLP approaches (San et al., 2023; Holt et al., 2024), as well as tailored sequential models like RNNs (Elman, 1990; Canzaniello et al., 2024), transformers (Vaswani et al., 2017), and pre-trained (Amad et al., 2025) or fine-tuned large language models (Makarov et al., 2025). Neural ODEs (Chen et al., 2018; Dupont et al., 2019; Alvarez et al., 2020) further extend deep learning capabilities to modelling continuous-time dynamics. Hybrid approaches (Raissi et al., 2019; Qian et al., 2021; Takeishi & Kalousis, 2021; Kuang et al., 2024) combine deep learning with mechanistic components, to improve sample efficiency and generalisation. Most existing DT frameworks seek to minimise transition errors across a training set, and, to our knowledge, no previous works optimise explicitly for decision-making over candidate policies.

## 5.2. Model-Based Reinforcement Learning

A DT is similar to a world model in model-based RL (MBRL) (Sutton, 1991), which is used to learn new policies. Some MBRL research has a similar motivation to ours, recognising that global model accuracy is unnecessary, and that optimising for this can be suboptimal for policy learning (Lambert et al., 2020; Grimm et al., 2020). Proposed solutions include policy-aware methods (Eysenbach et al., 2022; Wang et al., 2023; Ma et al., 2023) that favour areas of the state-action space that are traversed by the current policy, and value-aware methods (Farahmand et al., 2017; Farahmand, 2018; Grimm et al., 2020; Farquhar et al., 2021; Voelcker et al., 2022), which are more similar to $DT^2$, focusing model capacity in areas important to determine the current policy value. Nevertheless, these works operate in the online setting, where the true environment can be queried during training, whereas we focus on offline DT construction.

There are also similar works in offline MBRL which forego global model accuracy, typically to encode so-called 'pessimistic' dynamics to discourage the learned policy from traversing parts of the state-action space not covered by the offline dataset (Kidambi et al., 2020; Yu et al., 2020; Qiao et al., 2026). While useful for policy learning, such alterations to the learned model do not improve policy ranking abilities to the same extent as $DT^2$ (§6.2). Furthermore, since the models in MBRL are a means to learning a performant policy, maintaining simulation fidelity, for interpretability purposes, is not a priority, whereas this is essential for human-in-the-loop decision support with DTs.

## 5.3. Decision-Focused Learning

Our work is conceptually related to the field of decision-focused learning (DFL), which studies training predictive models whose outputs are used in downstream optimisation processes, with the objective of directly improving decision quality. Areas such as stochastic programming (Donti et al., 2017) and combinatorial optimisation (Wilder et al., 2019) have received attention in DFL, and extensions to learning models in MDPs have also been proposed (Futoma et al., 2020; Wang et al., 2021; Nikishin et al., 2022; Sharma et al., 2024). Generally, DFL requires some way to differentiate through the downstream optimisation process, which can limit the complexity of the models or policies considered in these works. Our method has minimal restrictions for the policies that can be considered in $\Pi$, and the DT model class, and we demonstrate good performance in a variety of complex, high-dimensional environments (§6.2). Furthermore, these works are largely designed to learn a model such that the optimal policy within it has high value, whereas we uniquely aim to learn a model such that policies from a candidate set are well ranked by it.

## 5.4. Off-Policy Evaluation

OPE methods are used to estimate $J(\pi_{\text{target}})$ from a dataset collected from some behavioural policy $\pi_{\text{behavioural}}$. Popular approaches include direct methods (Lagoudakis & Parr, 2003; Ernst et al., 2005; Munos & Szepesvári, 2008; Le et al., 2019), which learn and use $Q^{\pi_{\text{target}}}$ to estimate $J(\pi_{\text{target}})$, importance sampling methods (Precup et al., 2000; Hallak & Mannor, 2017; Liu et al., 2018; Xie et al., 2019; Nachum et al., 2019; Uehara et al., 2020; Zhang et al., 2020a;b), which use importance weights to correct for the difference in densities between $\pi_{\text{behavioural}}$ and $\pi_{\text{target}}$ to estimate $J(\pi_{\text{target}})$, and doubly robust methods (Jiang & Li, 2016; Thomas & Brunskill, 2016), which combine direct and importance sampling methods.

While OPE methods can be directly used for decision support, a primary limitation is their lack of interpretability and the difficulty in verifying their value estimates. DTs offer several advantages over such methods because they generate full simulations of each policy under consideration. Inspecting these simulations allows for the consideration of subjective criteria in decision-making, that may not be perfectly captured by a formally defined reward function. Furthermore, verification of DT value estimates is easier than for OPE methods, as if a simulation violates known constraints a user can choose to discard it.

## 6. Empirical Investigation

We now empirically validate the efficacy of DT². We cover the limited model capacity setting that relates to Theorems 3.1 and 3.3 in §6.1, before showing that their takeaways hold with expressive models as well in §6.2 and §6.3. Finally, we investigate the effects of $\lambda$ in $\mathcal{L}_{\text{DT}^2}$ in §6.4. We provide detailed experimental set-ups in Appendix B.

### 6.1. Restricted Hypothesis Spaces

To empirically demonstrate Theorems 3.1 and 3.3, we construct three toy scenarios which satisfy the required assumptions, where $\Phi \notin \mathcal{F}$ and $\mathcal{F}$ allows local improvement at $\theta^*$. N.B. in the following, we assume access to ground-truth policy values during DT² training, for simplicity. In the more realistic settings, from §6.2 onward, we demonstrate the full DT² pipeline, with FQE-derived proxy rankings.

### 6.1.1. LIMITED NEURAL CAPACITY

Consider a two-dimensional system $x_t = [x_{d,t}, x_{c,t}]^\top$ that decomposes into independent 'decoy' and 'critical' components, with an associated scalar action $a_t$. $x_t$ evolves according to $x_{d,t+1} = M \sin(x_{d,t})$ and $x_{c,t+1} = x_{c,t} + \epsilon a_t$, and we set $M$ and $\epsilon$ such that $\text{Var}(x_{d,t}) >> \text{Var}(x_{c,t})$. We set $\mathcal{F}$ as the family of three layer MLPs with a single-neuron bottleneck as the middle layer. Forcing an internal

one-dimensional representation prevents any model from simultaneously representing the independent evolution of $x_{d,t}$ and $x_{c,t}$, guaranteeing $\Phi \notin \mathcal{F}$.

We consider $\Pi$ with two constant-action policies, $\pi_{\text{high}} = 1.0$ and $\pi_{\text{low}} = 0.9$, and a reward $r(\mathbf{x}_t, a_t, \mathbf{x}_{t+1}) = x_{c,t}$ such that $\pi_{\text{high}} \succ \pi_{\text{low}}$. Across 20 seeds, we generate $\mathcal{D}$ by drawing $a_t \sim U[-1, 1]$, and train DTs with $\mathcal{L}_{\text{sim}}^{\text{MSE}}$ and $\mathcal{L}_{\text{DT}^2}$. Because $x_{d,t}$ dominates the variance in $\mathcal{D}$, the $\mathcal{L}_{\text{sim}}^{\text{MSE}}$-trained DT allocates it significant capacity, and consequently achieves only a 20% success rate across seeds in identifying $\pi_{\text{high}}$ as the optimal policy via model-estimated policy values. In contrast, $\mathcal{L}_{\text{DT}^2}$ encourages the bottlenecked model to identify that $x_c$ drives value differences, despite its small magnitude, resulting in 100% ranking accuracy.

### 6.1.2. MISSPECIFIED MECHANISTIC MODEL

We now consider a mechanistic setting, where the assumed functional form of the dynamics is incorrect. The true one-dimensional system has cubic action-dependent dynamics, $x_{t+1} = x_t + a_t - a_t^3 + \xi$, where $\xi \sim \mathcal{N}(0, 0.1)$, and we set $\mathcal{F}$ as the family of linear models, $x_{t+1} = x_t + \beta a_t$, such that $\Phi \notin \mathcal{F}$. We collect $\mathcal{D}$ by drawing $a_t \sim U[-1.5, 1.5]$.

We consider $\Pi$ containing three constant-action policies $\pi_1 = -0.58$, $\pi_2 = 0$, $\pi_3 = +0.58$, and a reward $r(x_t, a_t, x_{t+1}) = x_t$ such that $\pi_3 \succ \pi_2 \succ \pi_1$. We show the mechanistic models learnt by optimising $\mathcal{L}_{\text{sim}}^{\text{MSE}}$ and $\mathcal{L}_{\text{DT}^2}$, and the true dynamics in Figure 3. For the action range in $\mathcal{D}$, the cubic term $-a_t^3$ tends to dominate, creating a broadly negative correlation between action and state change $\delta = x_{t+1} - x_t$, and the $\mathcal{L}_{\text{sim}}^{\text{MSE}}$-trained DT learns this negative slope, consequently ordering $\pi_1 \stackrel{\cdot}{\succ} \pi_2 \stackrel{\cdot}{\succ} \pi_3$. DT², on the other hand, recognises that, in the local decision-critical region for $\Pi$, the linear term dominates, sacrificing global fit to learn a positive slope, recovering the true preference ordering.

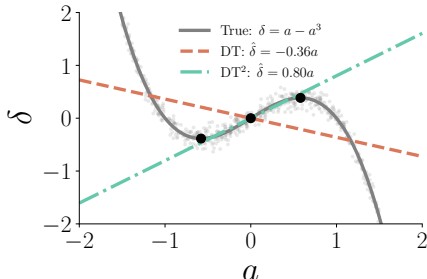

*Figure 3.* State delta across action values, as determined by mechanistic models learnt via $\mathcal{L}_{\text{sim}}^{\text{MSE}}$ and $\mathcal{L}_{\text{DT}^2}$, compared to the true $\Phi$. Black dots represent constant-action policies in $\Pi$.

### 6.1.3. PARTIALLY OBSERVED DYNAMICS

Finally, we investigate a system with partially observable dynamics, where an unobserved latent variable $z_t$ dictates

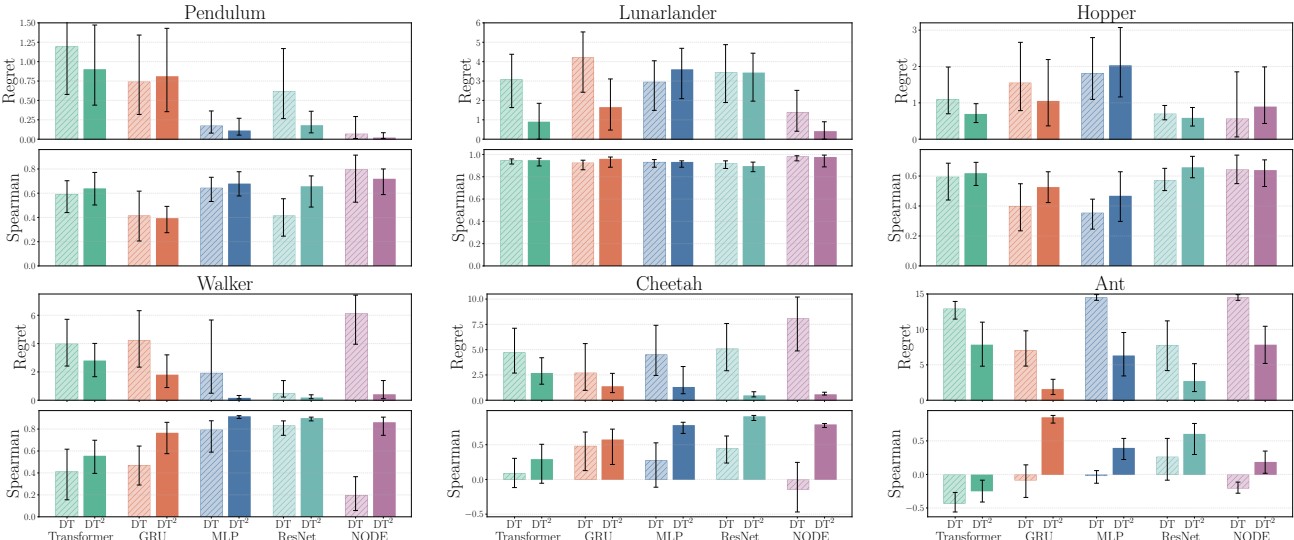

*Figure 4.* Regret and Spearman's correlation for preference orderings from $\mathcal{L}_{\text{sim}}^{\text{NLL}}$ (hashed) and $\mathcal{L}_{\text{DT}^2}$ (solid) DTs across base architectures in six continuous control environments. We report averages over 10 seeds, with 95% CIs.

the transition distribution, ensuring $\Phi \notin \mathcal{F}$ even with an expressive hypothesis space of three-layer MLPs. We consider a one-dimensional system governed by $x_{t+1} = x_t + a_t + z_t$, where $z_t \sim \mathcal{N}(0, 25)$ dominates, and set $\Pi$ with two constant-action policies, $\pi_{\text{high}} = 0.01$ and $\pi_{\text{low}} = -0.01$, and a reward $r(x_t, a_t, x_{t+1}) = x_t$ such that $\pi_{\text{high}} \succ \pi_{\text{low}}$. Across 20 seeds, we collect $\mathcal{D}$ by drawing $a_t \sim U[-1, 1]$.

Minimising $\mathcal{L}_{\text{sim}}^{\text{MSE}}$ encourages learning the conditional expectation $\mathbb{E}[x_{t+1} \mid x_t, a_t]$ by averaging over the unobserved latent $z_t$. While $a_t$ and $z_t$ are independent in the generative process, the high variance of $z_t$ can cause, with finite data, spurious correlations to emerge where actions coincide with opposing latent values. Consequently, the empirical conditional mean can exhibit a slope flip, causing the $\mathcal{L}_{\text{sim}}^{\text{MSE}}$ model to learn a negative relationship between $a_t$ and $x_{t+1}$ to fit the noise in $\mathcal{D}$. As such, it derives the correct preference ordering on $\Pi$ only 65% of the time. In contrast, $\text{DT}^2$ utilises the ranking loss to distinguish $\pi_{\text{high}}$ from $\pi_{\text{low}}$, better conserving the true positive slope of the action, achieving 100% ranking accuracy.

### 6.2. Expressive Hypothesis Spaces

We now wish to test our hypothesis that the takeaways from Theorems 3.1 and 3.3 persist even when $\mathcal{F}$ is expressive and could contain $\Phi$. We use six Mujoco (Todorov et al., 2012) and Gymnasium (Towers et al., 2025) continuous control environments (Pendulum, Lunarlander, Hopper, Walker, Cheetah, Ant) to act as 'real-world systems' $\Phi$ that span a range of complexities, from 4- to 35-dimensional state-action spaces. For each environment, we define a candidate set $\Pi$ of six policies taken from evenly-spaced checkpoints of a PPO (Schulman et al., 2017) training run. We

then generate $\mathcal{D}$ by deploying each $\pi \in \Pi$ in $\Phi$ for a fixed number of steps. To determine the ground-truth preference ordering over $\Pi$, we empirically estimate $J(\pi)$ for each $\pi \in \Pi$ using Monte Carlo rollouts in $\Phi$.

Firstly, we compare $\mathcal{L}_{\text{sim}}^{\text{NLL}}$-trained DTs, represents the prevailing context-agnostic approach, with training via $\text{DT}^2$. For $\text{DT}^2$, we first learn a $Q_\psi^\pi$ for each $\pi \in \Pi$ using FQE on $\mathcal{D}$, and then train the DT with $\lambda = 0.1$ and $\alpha = 1$. To show the generality of our method, we use a number of expressive architectures to form the basis of the DTs, training them to output the mean and variance of a multivariate Gaussian distribution for the next state. These include a transformer (Vaswani et al., 2017), gated recurrent unit (GRU) (Cho et al., 2014), multi-layer perceptron (MLP), residual network (ResNet) (He et al., 2016), and neural ODE (NODE) (Chen et al., 2018).

To determine the DT-derived preference orderings, $\hat{\succ}$, we also use Monte Carlo rollouts. To evaluate these against the ground-truth $\succ$, we report the Spearman's rank correlation (Spearman, 1904), to measure global ranking alignment, and the decision regret when selecting the optimal policy, to measure the value gap between $\pi^*$ and $\hat{\pi}^*$, in Figure 4. $\mathcal{L}_{\text{DT}^2}$ DTs outperform their $\mathcal{L}_{\text{sim}}^{\text{NLL}}$ counterpart in 83.3% of the architecture-environment-metric couplings, and there is no environment or architecture that $\text{DT}^2$ does not outperform in, on average. The average regret, across all environments and architectures, is 4.08 using $\mathcal{L}_{\text{sim}}^{\text{NLL}}$, and 1.88 using $\mathcal{L}_{\text{DT}^2}$, corresponding to a 54% reduction. Similarly, for Spearman's correlation, $\mathcal{L}_{\text{sim}}^{\text{NLL}}$ DTs average 0.45 while $\mathcal{L}_{\text{DT}^2}$ DTs average 0.66, corresponding to a 47% improvement.

In terms of simulation fidelity, we do see that there is a

*Table 1.* Regret and Spearman's correlation for preference orderings from different model learning paradigms in six continuous control environments. For $\mathcal{L}_{\mathrm{DT}^2}$ and $\mathcal{L}_{\mathrm{sim}}^{\mathrm{NLL}}$ we report the best-performing architectures from Figure 4 for each environment. We report averages over 10 seeds, with standard errors. **Red** indicates best performing, blue indicates second best.

| Method | Pendulum Regret ($\downarrow$) | Pendulum Spearman ($\uparrow$) | Lunarlander Regret ($\downarrow$) | Lunarlander Spearman ($\uparrow$) | Hopper Regret ($\downarrow$) | Hopper Spearman ($\uparrow$) | Walker Regret ($\downarrow$) | Walker Spearman ($\uparrow$) | Cheetah Regret ($\downarrow$) | Cheetah Spearman ($\uparrow$) | Ant Regret ($\downarrow$) | Ant Spearman ($\uparrow$) |
|---|---|---|---|---|---|---|---|---|---|---|---|---|
| $\mathcal{L}_{\mathrm{sim}}^{\mathrm{NLL}}$ | 0.08 (0.07) | **0.79** (0.11) | 1.41 (0.72) | **0.98** (0.01) | 0.59 (0.45) | 0.64 (0.06) | 0.55 (0.30) | 0.83 (0.01) | 2.77 (1.28) | 0.48 (0.16) | 7.10 (1.56) | −0.09 (0.16) |
| MOReL | 0.14 (0.05) | 0.69 (0.09) | 21.06 (9.77) | 0.73 (0.03) | **0.51** (0.17) | 0.57 (0.09) | 0.35 (0.06) | 0.87 (0.02) | 5.08 (1.20) | −0.29 (0.12) | 5.12 (1.23) | 0.22 (0.08) |
| MOPO | 0.36 (0.26) | 0.47 (0.06) | 1.45 (0.52) | 0.62 (0.09) | 8.50 (4.19) | −0.27 (0.12) | 8.13 (2.31) | 0.21 (0.15) | 9.57 (1.28) | −0.51 (0.12) | 11.98 (1.38) | 0.05 (0.13) |
| ROMI | 0.32 (0.27) | 0.49 (0.12) | 8.50 (1.34) | 0.80 (0.02) | 0.64 (0.11) | 0.60 (0.10) | 0.25 (0.07) | 0.84 (0.03) | 13.45 (0.34) | −0.87 (0.05) | 6.14 (2.45) | 0.33 (0.16) |
| VaGraM | 1.18 (0.79) | 0.50 (0.13) | 25.47 (15.28) | 0.84 (0.06) | 0.65 (0.15) | 0.46 (0.08) | 4.63 (2.28) | 0.43 (0.25) | 6.62 (2.00) | −0.26 (0.32) | 12.50 (0.89) | −0.38 (0.11) |
| HDTwin | 13.85 (6.55) | −0.10 (0.21) | 49.10 (39.01) | 0.54 (0.18) | 0.94 (0.49) | 0.40 (0.11) | 7.50 (1.15) | −0.13 (0.16) | 5.32 (2.04) | 0.16 (0.23) | 6.18 (2.07) | 0.24 (0.15) |
| FQE | **0.02** (0.01) | 0.76 (0.03) | 2.51 (1.85) | 0.83 (0.04) | 0.54 (0.27) | **0.78** (0.05) | 1.18 (0.66) | 0.78 (0.07) | 1.29 (0.30) | 0.84 (0.03) | **0** (0) | **0.94** (0) |
| $\mathcal{L}_{\mathrm{DT}^2}$ | 0.03 (0.02) | 0.72 (0.06) | **0.45** (0.45) | 0.97 (0.02) | 0.60 (0.16) | 0.66 (0.04) | **0.21** (0.08) | **0.91** (0.01) | **0.55** (0.16) | **0.90** (0.02) | 1.64 (0.68) | 0.84 (0.04) |

necessary trade-off to achieve this increased preference ordering performance. The average test set transition MSE using $\mathcal{L}_{\mathrm{sim}}^{\mathrm{NLL}}$ is 1.21, while for $\mathcal{L}_{\mathrm{DT}^2}$ it is 1.41, performing 17% worse in this regard. These results validate our core thesis: models optimised solely for simulation fidelity often misallocate capacity to dynamics irrelevant for policy ranking, even when using expressive hypothesis classes. Through explicit decision-targeted training, $\mathrm{DT}^2$ recovers significantly better preference orderings, at a modest MSE cost.

Secondly, we compare the preference orderings from the best performing $\mathcal{L}_{\mathrm{DT}^2}$ and $\mathcal{L}_{\mathrm{sim}}^{\mathrm{NLL}}$ models from Figure 4 with a handful of baseline models, including MBRL baselines, a recent DT baseline, and pure FQE rankings. The MBRL methods include MOReL (Kidambi et al., 2020) and MOPO (Yu et al., 2020), which both train neural ensembles of dynamics models and construct a 'pessimistic MDP' that penalises rewards in transitions where there is high model uncertainty, i.e. they redefine the reward as

$$\tilde{r}(\mathbf{x}_t, \mathbf{a}_t, \mathbf{x}_{t+1}) = r(\mathbf{x}_t, \mathbf{a}_t, \mathbf{x}_{t+1}) - \lambda\, u(\mathbf{x}_t, \mathbf{a}_t) \quad (18)$$

for some measure of model uncertainty $u$. We also compare with a more recent offline MBRL work, ROMI (Qiao et al., 2026), which alternates between training an ensemble of neural dynamics models and a SAC policy (Haarnoja et al., 2018), weighting the dynamics training using a value-aware approach to incorporate pessimism adversarially, rather than based on uncertainty. Finally, we adapt VaGraM (Voelcker et al., 2022), an online value-aware MBRL method, to be suitable for our offline setting, training using an MSE loss weighted by the average squared gradient norm of all $Q_\psi^\pi$-functions:

$$\mathbb{E}_{\mathcal{D}} \left[ \|\mathbf{x}' - \mu_\theta(\mathbf{x}, \mathbf{a})\|^2 \cdot \frac{1}{|\Pi|} \sum_{\pi \in \Pi} \|\nabla_{\mathbf{x}'} Q_\psi^\pi(\mathbf{x}', \pi(\mathbf{x}'))\|^2 \right].$$
$$(19)$$

This weights the importance of transitions by how sensitive candidate policy returns are to errors in those regions.

For the DT baseline, we compare with HDTwin (Holt et al., 2024), which performs an LLM-guided search process over hybrid DT structures (involving mechanistic and neural components), to find the architecture that minimises validation MSE.

These results are reported in Table 1. We see that $\mathrm{DT}^2$ tends to outperform these baselines, placing amongst the top two methods, in terms of both regret and Spearman's, in the majority of environments. FQE is the most competitive of the baselines, which validates the motivation for our method—to use the ranking abilities of OPE to supervise DT training—but, interestingly, we even see that $\mathrm{DT}^2$ improves upon its ranking supervisor in some environments.

While the MBRL methods go beyond optimising for raw simulation fidelity in ways that are effective for policy learning, we see that, generally, the preference orderings from these methods are about as competitive as the $\mathcal{L}_{\mathrm{sim}}^{\mathrm{NLL}}$ DTs. Clearly, bespoke alterations, such as via our $\mathrm{DT}^2$ method, are required to optimise dynamics models for the task of policy selection.

### 6.3. Case Study – Cancer Treatment Digital Twin

We now investigate a medical case study, where $\Pi$ is comprised of human-defined treatment plans. We use a `Cancer` environment from `DTR-Bench` (Luo et al., 2024) that simulates the progression of cancer with metastasis under chemotherapy and radiotherapy, based on the mathematical model of (Ghaffari et al., 2016), with a 9-dimensional state-action space and a reward function that balances tumour reduction against treatment toxicity. We define five distinct treatment plans to act as the candidate policy set: 1) no treatment, 2) fractionated radiotherapy, 3) metronomic chemotherapy, 4) adaptive combined therapy, and 5) aggressive combined therapy. We pursue a similar training and evaluation pipeline as in §6.2, using the ResNet architecture as the backbone for the $\mathcal{L}_{\mathrm{sim}}^{\mathrm{NLL}}$ and $\mathcal{L}_{\mathrm{DT}^2}$ DTs, since it tended to perform well in the previous environments.

We report the results in Table 2. Once again, using $\mathcal{L}_{\mathrm{DT}^2}$ leads to a significantly better regret and Spearman's rank correlation than $\mathcal{L}_{\mathrm{sim}}^{\mathrm{NLL}}$, indicating substantially better potential for clinical decision support. This is heightened by the fact that the MSE cost is only 2% here, meaning that decision-targeted training would minimally impact the result

*Table 2.* Regret and Spearman's correlation for preference orderings from $\mathcal{L}_{\text{sim}}^{\text{NLL}}$ and $\mathcal{L}_{\text{DT}^2}$ DTs in `Cancer` environment. Averaged over 5 seeds with standard errors.

| Loss | Regret ($\downarrow$) | Spearman ($\uparrow$) | MSE ($\downarrow$) |
|---|---|---|---|
| $\mathcal{L}_{\text{sim}}^{\text{NLL}}$ | 59.56 (0.83) | 0.10 (0.08) | 0.329 (0.01) |
| $\mathcal{L}_{\text{DT}^2}$ | 26.96 (11.04) | 0.64 (0.10) | 0.334 (0.01) |

of any human interrogation of DT-generated simulations in this case.

Furthermore, we test preference orderings across 11 policies that are unseen during training. These include six PPO checkpoints, obtained as in §6.2, and five further manually-defined policies: 1) pulsed chemotherapy, 2) hypofraction-ated radiotherapy, 3) aggressive combined therapy followed by maintenance chemo, 4) alternating modality treatment, and 5) dose-escalating combined treatment. In Table 3 we see that DT$^2$ performs best again, indicating that decision-targeting is not overfitting to the policies observed during training, but encourages learning dynamics that assist in general decision-making according to $r$. For some further analysis on unseen policy results across more environments, see Appendix C.

*Table 3.* Regret and Spearman's correlation for preference orderings from $\mathcal{L}_{\text{sim}}^{\text{NLL}}$ and $\mathcal{L}_{\text{DT}^2}$ DTs in `Cancer` environment across unseen policies. Averaged over 10 seeds with standard errors.

| Loss | Regret ($\downarrow$) | Spearman ($\uparrow$) |
|---|---|---|
| $\mathcal{L}_{\text{sim}}^{\text{NLL}}$ | 95.88 (18.23) | 0.40 (0.04) |
| $\mathcal{L}_{\text{DT}^2}$ | 49.27 (17.63) | 0.50 (0.06) |

### 6.4. The Effect of $\lambda$ in $\mathcal{L}_{DT^2}$

A key choice in $\mathcal{L}_{\text{DT}^2}$ is setting $\lambda$, which dictates where the model is placed on the trade-off between preserving the FQE-derived policy ranks and optimising for simulation fidelity. We investigate the performance of $\mathcal{L}_{\text{DT}^2}$ DTs across a range of values $\lambda \in [0, 0.75]$ in the six continuous control environments from §6.2, using the same set-ups and, again, using the ResNet architecture for each DT.

Figure 5 plots the Spearman's correlation and test MSE obtained across these $\lambda$ settings (N.B. that the $\lambda = 0$ setting is equivalent to standard training with $\mathcal{L}_{\text{sim}}^{\text{NLL}}$ only). As expected, we observe a consistent increase in ranking performance and decrease in simulation fidelity as $\lambda$ increases. Practitioners can therefore effectively use $\lambda$ to navigate to their desired point along this trade-off during training. However, we do see that ranking performance experiences diminishing returns at larger $\lambda$ values, and we experienced training instability with $\lambda \approx 1$, so we generally suggest setting a small $\lambda > 0$ for most use cases.

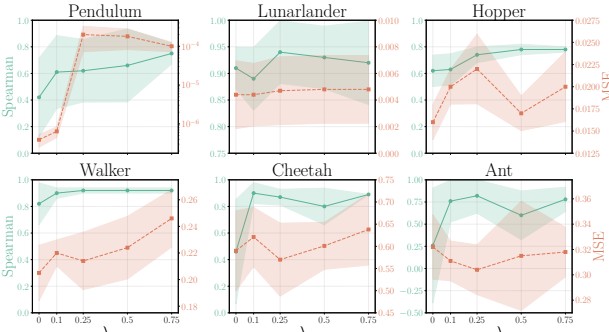

*Figure 5.* Spearman's correlation and test MSE for $\mathcal{L}_{\text{DT}^2}$ DTs in six continuous control environments across $\lambda$ levels. Averaged over 5 seeds, with 95% CIs.

## 7. Discussion

This work challenges the prevailing practice of maximising simulation fidelity in DT construction. We introduced DT$^2$, a training paradigm that aligns DTs with their primary downstream purpose: decision support. By distilling FQE-derived policy rankings into DT models, DT$^2$ induces significantly better preference orderings over candidate policies at a modest simulation fidelity cost. We demonstrate that these benefits hold across diverse environments, architectures and simulation losses (see further results using $\mathcal{L}_{\text{sim}}^{\text{MSE}}$ as the simulation loss in Appendix D), and that the trade-off between decision-making and simulation fidelity can be navigated via $\lambda$.

### 7.1. Limitations

DT$^2$ does have several limitations that are worth noting, and that can form the basis for various future works. The effect that DT$^2$ training has on the ranking ability of the resulting DT naturally depends on the quality of the proxy ranking targets used. Biased or high-variance FQE value estimates can occur if the state-visitation distributions of the candidate policies are poorly covered by $\mathcal{D}$, and these may be replicated by the trained DT. To address this, future work could focus on improving the general efficacy or sample-efficiency of OPE methods, or investigating some uncertainty-aware training extensions to DT$^2$, to neglect the signal from $\mathcal{L}_{\text{rank}}$ when the proxy values are likely poor. Additionally, DT$^2$ introduces computational overhead compared to standard training, requiring pre-trained $Q$-functions and $H$-step model rollouts during training. Potential strategies to mitigate this could involve using only a small representative subset of policies in $\Pi$ during training, or using adaptive bootstrapping horizons.

## Impact Statement

This paper presents work whose goal is to advance the field of machine learning. There are many potential societal

consequences of our work, none of which we feel must be specifically highlighted here.

## Acknowledgements

Harry Amad's studentship is funded by Canon Inc. This work was supported by Azure sponsorship credits granted by Microsoft's AI for Good Research Lab.

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

## A. Theorem Proofs

Herein we provide the proofs for Theorem 3.1 and Theorem 3.3.

### A.1. Proof of Theorem 3.1

*Proof.* By the theorem assumption, $\mathbb{E}_{(x,a)\sim\mathcal{D}}[D_{\mathrm{KL}}(\Phi\|\hat{\Phi}_{\theta^*})] > 0$. This implies the existence of a reachable state-action pair $(x_0, a_0) \in \mathrm{supp}(\mathcal{D})$ and a measurable set $G \subseteq \mathcal{X}$ where the transition probabilities differ:

$$\Phi(G \mid x_0, a_0) \neq \hat{\Phi}_{\theta^*}(G \mid x_0, a_0). \tag{20}$$

Let $p := \Phi(G \mid x_0, a_0)$ and $\hat{p} := \hat{\Phi}_{\theta^*}(G \mid x_0, a_0)$. Without loss of generality, assume $p > \hat{p}$.

Consider a decision scenario with horizon $H = 1$ (or, alternatively, where $r_t = 0$ for $t > 0$). Assume $|\mathcal{A}| \geq 2$, such that there be a second action $a_1 \in \mathcal{A}$ available at $x_0$. We define a bounded reward function $r : \mathcal{X} \times \mathcal{A} \times \mathcal{X} \to [0, 1]$:

$$r(x, a, x') := \mathbb{I}\{x = x_0\}\left(\mathbb{I}\{a = a_0\}\mathbb{I}\{x' \in G\} + \mathbb{I}\{a = a_1\}c\right), \tag{21}$$

where $c \in (0, 1)$ is a constant chosen such that $\hat{p} < c < p$.

Let $\Pi = \{\pi_0, \pi_1\}$ be a set of deterministic policies where $\pi_0(x_0) = a_0$ and $\pi_1(x_0) = a_1$. The policy values under the true system $\Phi$ are:

$$J(\pi_0) = p, \quad J(\pi_1) = c. \tag{22}$$

Since $p > c$, the true preference ordering is $\pi_0 \succ \pi_1$. However, under $\hat{\Phi}_{\theta^*}$ the estimated policy values are:

$$\hat{J}(\pi_0; \theta^*) = \hat{p}, \quad \hat{J}(\pi_1; \theta^*) = c. \tag{23}$$

Since $\hat{p} < c$, the model induces preference ordering $\pi_1 \hat{\succ} \pi_0$. Thus, $\hat{\Phi}_{\theta^*}$ yields an incorrect preference ordering. $\square$

### A.2. Proof of Theorem 3.3

To prove the existence of a better model in $\mathcal{F}$, we rely on the assumption of local improvement: that $\mathcal{F}$ contains an alternative model whose predictions on a local subset of transitions move in the correct direction relative to $\hat{\Phi}_{\theta^*}$.

*Proof.* From the proof of Theorem 3.1, we have $p > \hat{p}$ and we consider the same one-step decision task with reward

$$r(x, a, x') := \mathbb{I}\{x = x_0\}\left(\mathbb{I}\{a = a_0\}\mathbb{I}\{x' \in G\} + \mathbb{I}\{a = a_1\}c\right),$$

and policy set $\Pi = \{\pi_0, \pi_1\}$ where $\pi_0(x_0) = a_0$ and $\pi_1(x_0) = a_1$.

By assuming Definition 3.2, $\hat{p}' > \hat{p}$. Choose any constant

$$c \in (\hat{p}, \ \min\{p, \hat{p}'\}). \tag{24}$$

This interval is non-empty because $\hat{p}' > \hat{p}$ and $p > \hat{p}$.

**True ordering.**   As before, $J(\pi_0) = p$ and $J(\pi_1) = c$, hence $\pi_0 \succ \pi_1$ since $p > c$ by (24).

**Ordering under $\theta^*$.**   Under $\hat{\Phi}_{\theta^*}$, $\hat{J}(\pi_0; \theta^*) = \hat{p}$ and $\hat{J}(\pi_1; \theta^*) = c$, hence $\pi_1 \hat{\succ}_{\theta^*} \pi_0$ since $c > \hat{p}$.

**Ordering under $\theta'$.**   Under $\hat{\Phi}_{\theta'}$, $\hat{J}(\pi_0; \theta') = \hat{p}'$ and $\hat{J}(\pi_1; \theta') = c$, hence $\pi_0 \hat{\succ}_{\theta'} \pi_1$ since $\hat{p}' > c$ by (24).

Therefore, on the same decision problem $(r, \Pi)$, $\hat{\Phi}_{\theta^*}$ induces an incorrect preference ordering, whereas $\hat{\Phi}_{\theta'}$ induces the correct preference ordering. That is, $\hat{\Phi}_{\theta'}$ yields a strictly better preference ordering on $\Pi$ than $\hat{\Phi}_{\theta^*}$. $\square$

## B. Experimental Details

We provide detailed set-ups for all experiments run here. Furthermore, we will release code upon acceptance.

### B.1. Limited Neural Capacity Experiment

**System Dynamics.** The environment consists of a 2-dimensional state space $x_t = [x_{d,t}, x_{c,t}]^\top$ and a 1-dimensional action space $a_t \in \mathbb{R}$. The transitions are defined as:

$$x_{d,t+1} = M \sin(x_{d,t}) \tag{25}$$
$$x_{c,t+1} = x_{c,t} + \epsilon a_t \tag{26}$$

where $M = 500$, $\epsilon = 0.01$. The horizon is $H = 20$. The reward function is $r(x_t, a_t, x_{t+1}) = x_{c,t}$.

**Data Generation.** We collected a dataset $\mathcal{D}$ of 3,000 episodes using a behavior policy that samples actions uniformly $a \sim \mathcal{U}(-1, 1)$.

**Model Architecture.** We utilise an MLP with an explicit information bottleneck.

- **Input:** State (2) + Action (1).

- **Encoder:** Linear($3 \to 32$), Tanh, Linear($32 \to 1$).

- **Bottleneck:** The 1-dimensional output of the encoder represents the compressed state representation $z$.

- **Decoder:** Linear($2 \to 32$) (taking $z$ and $a$), Tanh, Linear($32 \to 2$).

**Training.** We trained for 30 epochs with a learning rate of $1 \times 10^{-3}$ using the Adam optimiser and $\mathcal{L}_{\text{sim}}^{\text{MSE}}$ as the simulation loss. For DT$^2$, we used $\lambda = 0.99$ and $\alpha = 1.0$. The ranking targets were computed using the ground truth value difference between $\pi_{\text{high}}$ and $\pi_{\text{low}}$.

### B.2. Misspecified Mechanistic Model Experiment

**System Dynamics.** The true system is a 1D scalar system governed by cubic control inputs:

$$x_{t+1} = x_t + (a_t - a_t^3) + \xi_t, \quad \xi_t \sim \mathcal{N}(0, 0.1) \tag{27}$$

**Hypothesis Space.** The model is constrained to be linear in actions:

$$\hat{x}_{t+1} = x_t + \beta a_t \tag{28}$$

Here, $\beta$ is the only learnable parameter.

**Training Data.** The dataset consists of 5,000 transitions where actions are sampled uniformly from: $a \sim \mathcal{U}(-1.5, 1.5)$.

**Policies.** We evaluate three constant policies:

1. $\pi_1 : a = -0.58$

2. $\pi_2 : a = 0.0$

3. $\pi_3 : a = +0.58$

The optimal policy is $\pi_3$.

**Training.** We trained for 2000 epochs with a learning rate of $0.01$ using the Adam optimiser and $\mathcal{L}_{\text{sim}}^{\text{MSE}}$ as the simulation loss. For DT$^2$, we used $\lambda = 0.9$ and $\alpha = 1.0$. The ranking targets were computed using the ground truth policy values.

## B.3. Partially Observed Dynamics Experiment

**System Dynamics.** The true system is a 1-dimensional linear system governed by an unobserved latent variable $z_t$ with high variance. The transitions are defined as:

$$x_{t+1} = x_t + a_t + z_t, \quad z_t \sim \mathcal{N}(0, 25) \tag{29}$$

where the state $x_t \sim \mathcal{N}(0, 1)$ and the action $a_t \in [-1, 1]$. The large standard deviation of the latent variable ($\sigma_z = 5$) relative to the action magnitude mimics a scenario where unobserved factors dominate the transition dynamics.

**Data Generation.** To emphasise data scarcity, we collected a small dataset $\mathcal{D}$ of $N = 100$ transitions. Actions in the dataset were sampled uniformly: $a \sim \mathcal{U}(-1, 1)$.

**Model Architecture.** We utilise an MLP with a single hidden layer to approximate the dynamics $x_{t+1} \approx \hat{\Phi}_\theta(x_t, a_t)$.

- **Input:** State $x_t$ and Action $a_t$ (Dimension: 2).

- **Hidden:** Linear($2 \to 16$), ReLU activation.

- **Output:** Linear($16 \to 1$) representing the predicted next state $\hat{x}_{t+1}$.

**Policies.** We evaluate the model's ability to rank two constant-action policies:

1. $\pi_{\text{high}} = 0.01$

2. $\pi_{\text{low}} = -0.01$

**Training.** We trained for 200 epochs with a learning rate of $1 \times 10^{-2}$ using the Adam optimiser. We used $\mathcal{L}_{\text{sim}}^{\text{MSE}}$ as the simulation loss. For DT$^2$, we used $\lambda = 0.9$ and a temperature $\alpha = 0.1$. The ranking targets were computed using the known ground truth policy values.

## B.4. Continuous Control Experiments

### B.4.1. ENVIRONMENTS

We use six continuous control environments from the Gymnasium library (Towers et al., 2025) (`https://github.com/Farama-Foundation/Gymnasium`, MIT license), utilising the MuJoCo physics engine (Todorov et al., 2012). These environments were selected to cover a diverse range of complexities, spanning state-action space dimensionalities from 4 to 35. The specific details for each environment are as follows:

- **Pendulum-v1** A classic control problem where the goal is to swing up and balance a pendulum in an inverted position from a random start.
    - $\mathcal{X}$: $d_{\mathcal{X}} = 3$, including the cosine and sine of the pendulum's angle, and its angular velocity.
    - $\mathcal{A}$: $d_{\mathcal{A}} = 1$, the scalar joint torque applied to the pendulum.

- **LunarLanderContinuous-v3**: A rocket trajectory optimisation task where the agent must safely land a spacecraft on a designated pad.
    - $\mathcal{X}$: $d_{\mathcal{X}} = 8$, including coordinates, linear velocities, angle, angular velocity, and boolean flags for leg ground contact.
    - $\mathcal{A}$: $d_{\mathcal{A}} = 2$, controls the main engine throttle and the side engine thrusters.

- **Hopper-v4**: A two-dimensional one-legged robot that aims to hop forward as fast as possible.
    - $\mathcal{X}$: $d_{\mathcal{X}} = 11$, comprising positional values and velocities of the body parts.
    - $\mathcal{A}$: $d_{\mathcal{A}} = 3$, controls the torques applied to the thigh, leg, and foot joints.

- **Walker2d-v5**: A two-dimensional bipedal robot that aims to walk forward.

- $\mathcal{X}$: $d_{\mathcal{X}} = 17$, including joint positions and velocities for both legs and the torso.
- $\mathcal{A}$: $d_{\mathcal{A}} = 6$, controls the torques for the joints.

- **HalfCheetah-v5**: A two-dimensional robot resembling a cheetah that aims to run forward.

  - $\mathcal{X}$: $d_{\mathcal{X}} = 17$, including positions and velocities of the torso and limbs.
  - $\mathcal{A}$: $d_{\mathcal{A}} = 6$, controls the torques on the forward and back thighs, shins, and feet.

- **Ant-v5**: A complex three-dimensional quadruped robot that aims to run forward.

  - $\mathcal{X}$: $d_{\mathcal{X}} = 27$, including the position, and orientation of the torso, joint angles, and velocities for all four legs.
  - $\mathcal{A}$: $d_{\mathcal{A}} = 8$, controls the torques on the hip and ankle joints of the four legs.

N.B. we run this environment with contact forces disabled, which would add an extra 78 state dimensions.

### B.4.2. POLICIES

To construct the set of candidate policies $\Pi$, we trained agents using the Proximal Policy Optimization (PPO) algorithm (Schulman et al., 2017), as implemented in the Stable Baselines3 library (Raffin et al., 2021) (`https://github.com/DLR-RM/stable-baselines3`, MIT license). We use the standard `MlpPolicy` architecture, and train for $T = 1 \times 10^6$ timesteps. The hyperparameters used during PPO training, across all environments, are mostly the default options within the Stable Baselines3 library, detailed in Table 4.

*Table 4.* PPO Hyperparameters used for training candidate policies.

| Hyperparameter | Value |
|---|---|
| Total Timesteps ($T$) | 1,000,000 (5,000,000 for Pendulum-v1) |
| Number of Environments ($N_{\text{envs}}$) | 8 |
| Steps per Environment ($N_{\text{steps}}$) | 2048 |
| Batch Size | 1024 |
| Learning Rate | $3 \times 10^{-4}$ |
| Discount Factor ($\gamma$) | 0.97 (0.95 for Pendulum-v1) |
| GAE Lambda ($\lambda$) | 0.95 |
| Clip Range ($\epsilon$) | 0.2 |
| Entropy Coefficient | 0.01 |
| Value Function Coefficient | 0.5 |
| Optimizer | Adam |

We constructed the candidate policy set $\Pi$ by saving checkpoints at evenly spaced intervals during training. Specifically, we selected policies at $\{0, 0.2T, 0.4T, 0.6T, 0.8T, T\}$ steps. The ground-truth preference ordering over $\Pi$, denoted by $\succ$, was established by estimating the expected discounted return $J(\pi)$ for each $\pi \in \Pi$. This was calculated via Monte Carlo estimation using 500 rollouts in the true environment.

To collect the DT training dataset, $\mathcal{D}$, we unroll each $\pi \in \Pi$ in the true environments for a $N$ steps. For `Pendulum` we set $N = 5000$, for `Lunarlander` we set $N = 1000$, and for the remaining environments we set $N = 10000$.

### B.4.3. FITTED Q EVALUATION

To obtain the proxy ground-truth rankings for decision-targeted training, we estimated the action-value function $Q^{\pi}(s, a)$ for each policy $\pi \in \Pi$ using Fitted Q Evaluation (FQE) (Le et al., 2019). $Q_{\psi}^{\pi}$ was parametrised as an MLP with Layer Normalization and SiLU activations. The network takes the concatenated state and action vectors $(s, a)$ as input and outputs a scalar Q-value.

To handle varying reward scales across environments and improve training stability, we standardised the rewards in the offline dataset to have zero mean and unit variance. $Q_{\psi}^{\pi}$ was trained to predict the cumulative return of these standardised rewards. During inference, predictions were rescaled to the original magnitude.

$Q_\psi^\pi$ were trained by minimising the expected Bellman error over the offline dataset $\mathcal{D}$. We employed a target network $Q_{\bar{\psi}}^\pi$ with an identical architecture, whose weights were updated via a hard update every 5 epochs.

We trained a separate $Q_\psi^\pi$ for each policy in $\Pi$ using the AdamW optimizer, with gradient norm clipping. The specific hyperparameters are summarised in Table 5.

*Table 5.* Hyperparameters for Fitted Q Evaluation.

| Hyperparameter | Value |
| --- | --- |
| Hidden Dimensions | $[256, 256]$ |
| Activation Function | SiLU |
| Normalization | LayerNorm |
| Training Epochs | 200 |
| Batch Size | $1,024$ |
| Learning Rate | $3 \times 10^{-4}$ |
| Target Update Frequency | Every 5 epochs |
| Target Action Samples ($K$) | 32 |
| Optimizer | AdamW |
| Gradient Clipping Norm | 10.0 |

### B.4.4. DIGITAL TWIN TRAINING

We evaluated our method using five distinct backbone architectures to demonstrate the architecture-agnostic nature of DT$^2$. All dynamics models were implemented as probabilistic ensembles (though we treat them as single stochastic models for the purpose of this work) that output the mean $\mu_\theta(\mathbf{x}, \mathbf{a})$ and diagonal log-variance $\log \Sigma_\theta(\mathbf{x}, \mathbf{a})$ of a Gaussian distribution $\mathcal{N}(\mu_\theta, \Sigma_\theta)$.

To improve training stability, we standardised inputs (state and action) to zero mean and unit variance using statistics computed from the offline dataset. The models were trained to predict the normalised state difference $\Delta \mathbf{x} = \mathbf{x}_{t+1} - \mathbf{x}_t$. The final prediction is obtained by denormalising the output and adding it to the current state.

To ensure numerical stability, the predicted log-variance is clamped to the range $[-10.0, 2.0]$. Additionally, during trajectory generation, the predicted next states are explicitly clipped to the specific physical bounds of the environment (e.g., joint limits) to ensure the simulation remains valid.

**Base Architectures.** We evaluated our method using five distinct feature extraction backbones. The output of each backbone is projected by two separate linear layers to produce the mean and log-variance vectors. The backbone details are as follows:

- **MLP**: A standard feed-forward network. It projects the input to the hidden dimension, followed by a SiLU activation, a second linear layer, and another SiLU activation.

- **ResNet**: A residual network consisting of an input projection followed by 4 residual blocks. Each block applies: Linear $\rightarrow$ LayerNorm $\rightarrow$ SiLU $\rightarrow$ Dropout(0.1) $\rightarrow$ Linear $\rightarrow$ LayerNorm, with a skip connection adding the block input to the output. A final SiLU activation is applied after the blocks.

- **Neural ODE (NODE)**: This backbone models the state evolution as an Ordinary Differential Equation (Chen et al., 2018). The hidden state transformation is defined as $\mathbf{z}(T) = \text{ODESolve}(\mathbf{z}(0), f, 0, T)$, where $f$ is a neural network (Linear $\rightarrow$ Tanh $\rightarrow$ Linear) representing the derivative $d\mathbf{z}/dt$. We used an explicit Runge-Kutta 4 (RK4) solver with a fixed step size ($N = 1$ step).

- **Transformer**: Utilizes a standard `TransformerEncoder` (Vaswani et al., 2017) with 2 layers, 4 attention heads, and a feed-forward dimension of twice the hidden size. To capture temporal dependencies, the model receives a context window of the $T = 8$ most recent transitions. Learned positional embeddings are added to the input sequence.

- **GRU**: A Gated Recurrent Unit network (Cho et al., 2014) with 2 stacked layers. Similar to the Transformer, it processes a history sequence of length $T = 8$ to predict the next state.

**Training Hyperparameters.** All models were trained using the AdamW optimizer with gradient clipping. We employed early stopping based on the validation loss to prevent overfitting. The full set of hyperparameters is listed in Table 6.

*Table 6.* Hyperparameters for Digital Twin Training.

| Hyperparameter | Standard DT | DT$^2$ |
|---|---|---|
| Optimizer | AdamW | AdamW |
| Learning Rate | $3 \times 10^{-4}$ | $3 \times 10^{-4}$ |
| Batch Size | 1024 | 1024 |
| Hidden Dimension | 64 | 64 |
| Gradient Clipping Norm | 10.0 | 10.0 |
| Max Epochs | 2000 | 2000 |
| Early Stopping Patience | 20 epochs | 20 epochs |
| *DT$^2$ Specific Parameters* | | |
| Ranking Weight ($\lambda$) | – | 0.1 |
| Rank Loss Temperature ($\alpha$) | – | 1.0 |
| Rollout Horizon ($H$) | – | 20 |
| Rollout Episodes ($M$) | – | 128 |
| Bootstrapping | – | Yes (using FQE) |

N.B. in the `Ant` environment, we set $H = 50$. For DT$^2$ and standard DTs, we used $\mathcal{L}_{\text{sim}}^{\text{NLL}}$ as the simulation loss for the results displayed in §6.2, and results using $\mathcal{L}_{\text{sim}}^{\text{MSE}}$ are in Appendix D.

**VaGraM Baseline.** We adapted the VaGraM method (Voelcker et al., 2022) to our offline ranking setting. Originally designed for online MBRL, alongside learning a single policy, we adapt it to train on batched data, and to weight transition errors based on the sensitivity of the value functions of *all* candidate policies in $\Pi$.

The model was trained to minimise a weighted Mean Squared Error (MSE) objective:

$$\mathcal{L}_{\text{VaGraM}}(\theta) = \mathbb{E}_{(\mathbf{x}, \mathbf{a}, \mathbf{x}') \sim \mathcal{D}} \left[ ||\mathbf{x}' - \mu_\theta(\mathbf{x}, \mathbf{a})||^2 \cdot w(\mathbf{x}') \right], \tag{30}$$

where the weight $w(\mathbf{x}')$ is the average squared norm of the value gradients at the next state across all policies:

$$w(\mathbf{x}') = \frac{1}{|\Pi|} \sum_{\pi \in \Pi} ||\nabla_{\mathbf{x}'} Q_\psi^\pi(\mathbf{x}', \pi(\mathbf{x}'))||_2^2. \tag{31}$$

The gradients $\nabla_{\mathbf{x}'} Q$ were computed via backpropagation through the pre-trained FQE networks (which were frozen during this process). We trained the VaGraM model using the MLP backbone and the same settings as the standard DTs.

**MOReL Baseline** We implement the MOReL (Kidambi et al., 2020) dynamics ensemble following the parameterisation described in the original work. Each ensemble member is a Gaussian mean model of the form

$$\hat{x}_{t+1} = x_t + \sigma_\delta \cdot \text{MLP}\left( \frac{x_t - \mu_s}{\sigma_s}, \frac{a_t - \mu_a}{\sigma_a} \right), \tag{32}$$

where $\mu_s, \sigma_s, \mu_a, \sigma_a$ are the empirical mean and standard deviation of states and actions in $\mathcal{D}$, and $\sigma_\delta$ is the empirical standard deviation of one-step state deltas. Each model is trained by minimising the MSE on normalised deltas, equivalent to Gaussian MLE under fixed diagonal covariance. The specific architecture and training hyperparameters are given in Table 7.

After training, we compute the pairwise $\ell_2$ disagreement between all ensemble members on the training transitions and set the Unknown State-Action Detector (USAD) threshold as

$$\tau = \mu_d + \beta \cdot \sigma_d, \tag{33}$$

where $\mu_d$ and $\sigma_d$ are the mean and standard deviation of the per-transition disagreement scores, and $\beta = 5$ following the paper default. Transitions flagged as unknown by USAD receive the halt reward $r_{\text{halt}} = r_{\text{min}} - \delta_r$, where $r_{\text{min}}$ is the minimum reward observed in $\mathcal{D}$ and $\delta_r$ is an environment-specific offset. For policy evaluation, we roll out each candidate policy $\pi \in \Pi$ in the resulting pessimistic MDP.

*Table 7.* Hyperparameters for the MOReL dynamics ensemble.

| Hyperparameter | Value |
| --- | --- |
| Ensemble Size | 4 |
| Hidden Dimension | 200 |
| Hidden Layers | 4 |
| Activation | ReLU |
| Max Epochs | 2,000 |
| Early Stopping Patience | 20 epochs |
| Batch Size | 256 |
| Learning Rate | $5 \times 10^{-4}$ |
| Optimizer | Adam |
| Bootstrap Sampling | Yes |
| Validation Fraction | 0.1 |
| Threshold $\beta$ | 5.0 |

**MOPO Baseline** We implement the MOPO (Yu et al., 2020) dynamics ensemble following the architecture described in the original work. Each ensemble member is a probabilistic model that outputs the mean and diagonal log-variance of a Gaussian distribution over normalised state deltas:

$$p_\theta(\Delta x \mid x_t, a_t) = \mathcal{N}\big(\mu_\theta(x_t, a_t), \, \mathrm{diag}(\sigma_\theta^2(x_t, a_t))\big), \tag{34}$$

where $\Delta x = x_{t+1} - x_t$. Inputs are normalised to zero mean and unit variance using statistics computed from $\mathcal{D}$, and outputs are denormalised prior to rollout. Spectral normalisation is applied to all linear layers except the log-variance head, following the original implementation. Log-variances are soft-clamped to the range $[\log \sigma_{\min}^2, \log \sigma_{\max}^2]$ via learnable bound parameters. Each member is trained by minimising the Gaussian NLL with a small regularisation term penalising the spread of the log-variance bounds.

We train an ensemble of 7 members and retain the 5 with the lowest holdout NLL as the elite subset. Policy evaluation uses only the elite models.

The per-transition uncertainty is defined as the maximum across elite members of the $\ell_2$ norm of the predicted standard deviation:

$$u(x_t, a_t) = \max_{e \in \mathcal{E}} \left\| \sigma_\theta^{(e)}(x_t, a_t) \right\|_2, \tag{35}$$

where $\mathcal{E}$ denotes the elite set. The penalised reward used for policy evaluation is then $\tilde{r}(x_t, a_t) = r(x_t, a_t) - \lambda \cdot u(x_t, a_t)$, where $\lambda$ is the penalty coefficient. The specific architecture and training hyperparameters are given in Table 8.

*Table 8.* Hyperparameters for the MOPO dynamics ensemble.

| Hyperparameter | Value |
| --- | --- |
| Ensemble Size | 7 |
| Elite Size | 5 |
| Hidden Dimension | 200 |
| Hidden Layers | 4 |
| Activation | SiLU |
| Spectral Normalisation | Yes (trunk & mean head) |
| Max Epochs | 2,000 |
| Early Stopping Patience | 20 epochs |
| Elite Holdout Size | 1,000 transitions |
| Batch Size | 256 |
| Learning Rate | $3 \times 10^{-4}$ |
| Optimizer | Adam |
| Bootstrap Sampling | Yes |

**ROMI Baseline**   We implement ROMI (Qiao et al., 2026), which jointly learns a dynamics ensemble, an adaptive sample-reweighting network, and a SAC policy. Each ensemble member is a probabilistic model with the same Gaussian parameterisation over normalised state deltas as MOPO, but without spectral normalisation. The ensemble is first pre-trained for 50 epochs by minimising the standard Gaussian NLL on bootstrapped subsets of $\mathcal{D}$.

Following the pre-training phase, ROMI alternates between (i) SAC policy updates on mixed real and model data, and (ii) a bilevel optimisation that jointly refines the dynamics and an adaptive weighting network $w_\nu(x, a, x')$. The weighting network is an MLP that outputs per-transition weights in the range $[w_{\min}, w_{\max}]$ via a $\tanh$ output activation. The inner level of the bilevel objective trains dynamics member $\Phi_\theta^{(e)}$ with a weighted NLL:

$$\mathcal{L}_{\text{inner}} = \mathbb{E}_{(x,a,x') \sim \mathcal{D}} \left[ w_\nu(x, a, x') \cdot \ell_{\text{NLL}}(x, a, x'; \theta) \right], \tag{36}$$

where $w_\nu$ is held fixed. The outer level then updates $\nu$ to minimise a robust value-aware loss. Specifically, given a one-step look-ahead from the updated dynamics $\hat{x}' = \mu_{\theta+}(x, a)$, and a pessimistic value target computed by perturbing $x'$ with Gaussian noise of scale $\sigma_u$ and taking the minimum $Q$-value over $K$ samples:

$$V^-(x') = \min_{k=1}^{K} \min(Q_1(x' + \epsilon_k, \pi(x' + \epsilon_k)), Q_2(x' + \epsilon_k, \pi(x' + \epsilon_k))), \quad \epsilon_k \sim \mathcal{N}(0, \sigma_u^2 I), \tag{37}$$

the outer loss encourages the dynamics to predict next states whose value matches this robust target:

$$\mathcal{L}_{\text{outer}} = \left( V(\hat{x}') - V^-(x') \right)^2. \tag{38}$$

The SAC policy is trained on batches mixing real transitions (ratio $\rho = 0.5$) with model rollouts of horizon $H = 5$ stored in a replay buffer.

Following the ROMI paper, we use $\sigma_u = 0.1$ across all environments, except Hopper ($\sigma_u = 1.0$) and Walker ($\sigma_u = 0.01$).

The full set of hyperparameters is given in Table 9.

**HDTwin Baseline**   We implement HDTwin (Holt et al., 2024) using the official repository here: `https://github.com/samholt/HDTwinGen` (MIT License). This method prompts an LLM to propose a Python class that represents the functional form of a DT, which is then trained on the training data. Per-variable validation MSE is then calculated, and reported back to the LLM, which is prompted to suggest changes to improve this validation MSE. The functional form is then iteratively improved in this manner. We use `GPT-5` via the Azure API for the LLM, allow 3 cycles of iterative improvement, and train each model for up to 2000 epochs with early stopping and patience of 20. All other hyperparameters are kept at default.

## B.5. Cancer Experiment

### B.5.1. ENVIRONMENT DETAILS

We used the `GhaffariCancerEnv` from `DTR-Bench`/`DTRGym` (Luo et al., 2024) (`https://github.com/GilesLuo/DTR-Bench`, `https://github.com/GilesLuo/DTRGym`, MIT License), which simulates the treatment of cancer with metastasis under combined radiotherapy and chemotherapy. The environment is based on the mathematical model by Ghaffari et al. (2016).

- $\mathcal{X}$: $d_\mathcal{X} = 7$, log-transformed cell counts and drug concentrations.

- $\mathcal{A}$: $d_\mathcal{A} = 2$, radiotherapy dose ($D \in [0, 10]$ Gy), chemotherapy concentration ($v_M \in [0, 8]$).

- We use 'Setting 5' of the environment, representing a challenging medical scenario. This includes:

    - State transition noise ($\sigma_{\text{state}} = 0.5$).
    - Observation noise ($\sigma_{\text{obs}} = 0.2$).
    - Pharmacokinetic/Pharmacodynamic parameter noise ($\sigma_{\text{pkpd}} = 0.1$).
    - A 50% probability of missing observations at any given timestep.

*Table 9.* Hyperparameters for ROMI.

| Hyperparameter | Value |
| --- | --- |
| *Dynamics Ensemble* | |
| Ensemble Size | 4 |
| Hidden Dimension | |
| Hidden Layers | 2 |
| Activation | SiLU |
| Pre-train Epochs | 50 |
| Dynamics Learning Rate | $3 \times 10^{-4}$ |
| Bootstrap Sampling | Yes |
| *Weighting Network* | |
| Hidden Dimension | 64 |
| Hidden Layers | 2 |
| Weight Range $[w_{\min}, w_{\max}]$ | $[0.5, 2.0]$ |
| Weight Learning Rate | $1 \times 10^{-4}$ |
| Bilevel Inner Learning Rate | $3 \times 10^{-4}$ |
| Adversarial Train Steps | 1,000 |
| Uncertainty Samples $(K)$ | 20 |
| *SAC Policy* | |
| Actor/Critic Hidden Dimension | 256 |
| Actor Learning Rate | $1 \times 10^{-4}$ |
| Critic Learning Rate | $3 \times 10^{-4}$ |
| Soft Update $(\tau)$ | $5 \times 10^{-3}$ |
| *Training Schedule* | |
| Epochs | 500 |
| Batch Size | 256 |
| Real Data Ratio $(\rho)$ | 0.5 |
| Policy Updates per Epoch | 200 |
| Rollout Horizon $(H)$ | 5 |
| Rollout Batch Size | 512 |

### B.5.2. TRAINING POLICIES

For this experiment, we defined 5 interpretable clinical policies to construct $\Pi_{\text{train}}$:

1. **No Treatment**: Baseline policy with zero intervention ($D = 0, v_M = 0$).

2. **Fractionated Radiotherapy**: 2 Gy/day radiation, 5 days/week, with weekends off. No chemotherapy.

3. **Metronomic Chemotherapy**: Continuous low-dose chemotherapy ($v_M = 2.0$) administered daily. No radiotherapy.

4. **Adaptive Combined Therapy**: A reactive strategy that monitors tumour burden. If the log-size exceeds a threshold (50% of initial burden), it applies aggressive combination therapy ($D = 2.0, v_M = 4.0$). Otherwise, no treatment is given.

5. **Aggressive Combined Therapy**: High dose of both modalities ($D = 4.0, v_M = 6.0$) administered daily.

### B.5.3. EVALUATION POLICIES

To test the generalisation capabilities of DT$^2$, we evaluated the models on a set of 11 unseen policies, including 6 PPO checkpoints along a training run of 200,000 steps, and the 5 following manually-defined policies:

- **Pulsed Chemotherapy**: High dose chemotherapy ($v_M = 5.0$) administered once every 21 days.

- **Hypofractionated Radiotherapy**: High radiation dose ($D = 8.0$) administered every 3 days.

- **Induction-Maintenance**: Aggressive combination therapy ($D = 2.0, v_M = 4.0$) for the first 30 days, followed by low-dose maintenance chemotherapy ($v_M = 1.5$).

- **Alternating Modality**: A 14-day cycle consisting of 7 days of daily radiotherapy ($D = 2.0$) followed by 7 days of daily chemotherapy ($v_M = 3.0$).

- **Dose Escalation**: Doses for both modalities linearly ramp up from 0 to a maximum ($D = 3.0, v_M = 5.0$) over the first 100 days.

### B.5.4. DIGITAL TWIN TRAINING

The training pipeline followed the same structure as the continuous control experiments, with the following environment-specific adjustments:

- **Dataset**: We collected 1,000 steps from each of the 5 training expert policies.

- **Hyperparameters**: We used a batch size of 512 and a discount factor $\gamma = 0.99$. The Q-networks were trained for 500 epochs.

### B.6. Lambda Ablation

For this experiment, we use the ResNet backbone in all models. All settings are kept the same as in the continuous control experiments, except we vary $\lambda \in \{0, 0.1, 0.25, 0.5, 0.75\}$.

## C. Unseen Policy Rankings in Continuous Control Environments

We now report some further results to assess how $DT^2$ performs in ranking policies that are unseen during training. Here, we compare $\mathcal{L}_{\text{sim}}^{\text{NLL}}$ and $\mathcal{L}_{DT^2}$ in ranking unseen policies across the three smaller continuous control environments from §6.2 (`Pendulum`, `Lunarlander`, `Hopper`). Again, we use the ResNet architecture, for consistency with §6.3.

In this case, since it is not so clear how 'expert defined' policies might look, we investigate rankings of five simple policies that are not seen during training, that involve taking 1) random actions, 2) constant **0** action, 3) constant minimum action, 4) constant median action, and 5) constant maximum action. These policies are generally much less sophisticated, and worse-performing, than the PPO policies used to collect the datasets and train $DT^2$ in §6.2, so they can be considered quite out of distribution.

In Table 10 we report the ranking performance of $\mathcal{L}_{\text{sim}}^{\text{NLL}}$ and $\mathcal{L}_{DT^2}$ DTs across these simple policies. We see that $DT^2$ training still leads to an increase in ranking performance, although, as expected, to a lesser extent than when ranking policies that are seen during training.

*Table 10.* Regret and Spearman's correlation on preference orderings of five unseen constant-action policies in continuous control environments. We report averages over 5 seeds, with standard errors.

| Loss | Pendulum | | LunarLander | | Hopper | |
|---|---|---|---|---|---|---|
| | Regret ($\downarrow$) | Spearman ($\uparrow$) | Regret ($\downarrow$) | Spearman ($\uparrow$) | Regret ($\downarrow$) | Spearman ($\uparrow$) |
| $\mathcal{L}_{\text{sim}}^{\text{NLL}}$ | 8.70 (6.03) | 0.340 (0.201) | 211.78 (83.92) | 0.760 (0.058) | 1.83 (0.91) | 0.811 (0.042) |
| $\mathcal{L}_{DT^2}$ | 5.37 (4.35) | 0.580 (0.169) | 102.15 (63.56) | 0.780 (0.061) | 0.61 (0.60) | 0.911 (0.042) |

# D. Comparisons with MSE Simulation Loss

We display results for the continuous control experiments using $\mathcal{L}_{\text{sim}}^{\text{MSE}}$ for the simulation loss, rather than $\mathcal{L}_{\text{sim}}^{\text{NLL}}$ in Figure 6. We see similar takeaways from these results as those in §6.2. Namely, DT$^2$ outperforms its standard counterpart in 76.7% of the environment-architecture-metric couplings. In general, the DTs using $\mathcal{L}_{\text{sim}}^{\text{MSE}}$ as their simulation loss seem to perform worse than when using $\mathcal{L}_{\text{sim}}^{\text{NLL}}$.

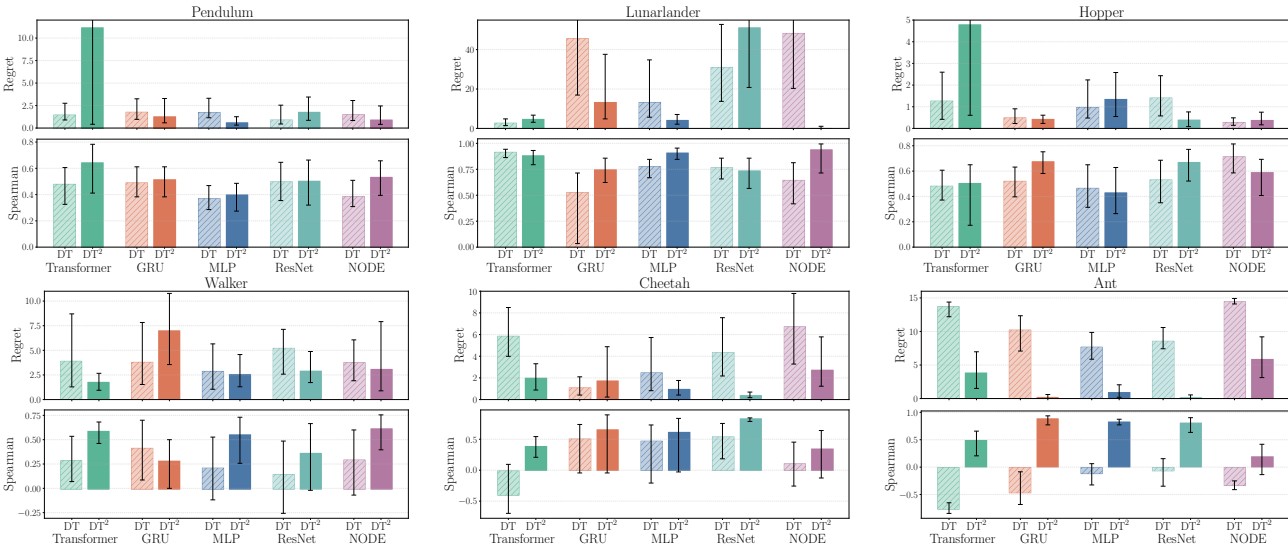

*Figure 6.* Regret and Spearman's correlation for preference orderings from $\mathcal{L}_{\text{sim}}^{\text{MSE}}$ (hashed) and $\mathcal{L}_{\text{DT}^2}$ (solid) DTs across base architectures in six continuous control environments. We report averages over 10 seeds, with 95% CIs. For visual purposes, the top end of some error bars are cropped out.

# E. Ranking Loss Ablation

We now consider some variants to the differentiable ranking loss used for DT$^2$. Let $\mathbf{y} \in \mathbb{R}^M$ be the vector of target proxy values (derived from FQE) for the $M$ policies in a batch, and $\hat{\mathbf{y}} \in \mathbb{R}^M$ be the corresponding vector of cumulative returns predicted by the DT rollouts. Let $\mathcal{C} = \{(i,j) \mid 1 \leq i < j \leq M\}$ denote the set of all unique pairs in the batch.

**Pairwise Hinge Loss.** This variant enforces that the DT's predicted value difference between two policies preserves the sign of the ground-truth difference, with a sufficient margin. To ensure the margin scales appropriately across different environments with varying reward magnitudes, we implemented an adaptive margin $\epsilon$ based on the mean absolute value of the targets:

$$\epsilon = 0.1 \cdot \frac{1}{M} \sum_{k=1}^{M} |y_k|. \tag{39}$$

The loss is defined as:

$$\mathcal{L}_{\text{Hinge}}(\theta) = \frac{1}{|\mathcal{C}|} \sum_{(i,j) \in \mathcal{C}} \max\left(0, \epsilon - \text{sign}(y_i - y_j)(\hat{y}_i - \hat{y}_j)\right). \tag{40}$$

This penalises the model if the predicted ordering is incorrect or if the separation between the pair is smaller than $\epsilon$.

**ListNet Loss.** ListNet (Cao et al., 2007) treats the ranking problem as a probability distribution matching problem. It maps the vector of values to a probability distribution using a softmax function.

Let $P_y$ and $P_{\hat{y}}$ be the softmax distributions of the target and predicted values, respectively, controlled by a temperature parameter $\alpha$:

$$P_y(\pi_i) = \frac{\exp(y_i/\alpha)}{\sum_{k=1}^{M} \exp(y_k/\alpha)}, \quad P_{\hat{y}}(\pi_i) = \frac{\exp(\hat{y}_i/\alpha)}{\sum_{k=1}^{M} \exp(\hat{y}_k/\alpha)}. \tag{41}$$

The loss is calculated as the Cross-Entropy between these distributions:

$$\mathcal{L}_{\text{ListNet}}(\theta) = -\sum_{i=1}^{M} P_y(\pi_i) \log\left(P_{\hat{y}}(\pi_i)\right). \tag{42}$$

Unlike the pairwise approaches, ListNet considers the entire list of policies simultaneously.

## E.1. Continuous Control Results

We display results for the continuous control experiments with these variants in Figure 7, and report their average performances in Table 11. Across all environment-architecture-metric couplings, our smoothed Kendall formulation has the highest win-rate. While the other ranking formulations can improve upon $\mathcal{L}_{\text{sim}}^{\text{NLL}}$, our proposed method tends to work best.

In Table 11, we can see that our smoothed Kendall formulation achieves the lowest average regret, and the highest average Spearman's correlation. Interestingly, in this case, while the Hinge and ListNet formulations improve on $\mathcal{L}_{\text{sim}}^{\text{NLL}}$ in terms of average Spearman's, they are worse in terms of average regret. As we can see from Figure 7, they each have some cases where they have a very poor regrets (e.g. in the `Pendulum` and `Lunarlander` environments), which drives up their averages for this metric.

*Table 11.* Average Spearman's correlation and regret for different DT variants across all architectures and continuous control task.

| | Loss | Regret ($\downarrow$) | Spearman ($\uparrow$) |
|---|---|---|---|
| | $\mathcal{L}_{\text{sim}}^{\text{NLL}}$ | 4.08 | 0.45 |
| $\mathcal{L}_{\text{DT}^2}$ | Kendall | 1.88 | 0.66 |
| | Hinge | 4.57 | 0.59 |
| | ListNet | 11.51 | 0.53 |

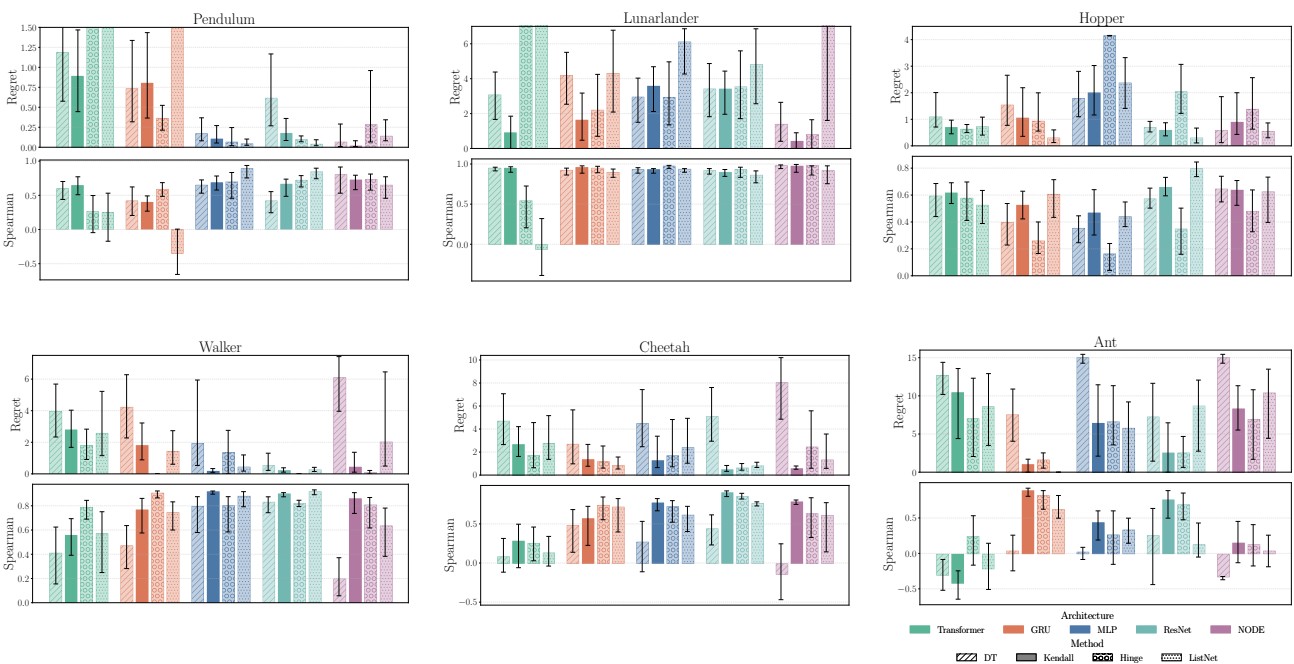

*Figure 7.* Regret and Spearman's correlation for preference orderings from DTs trained via $\mathcal{L}_{\text{sim}}^{\text{NLL}}$ DTs (hashed) and $\mathcal{L}_{\text{DT}^2}$ with ranking losses of our smoothed Kendall's (solid), a hinge loss (circles), and a ListNet loss (dotted) across different base architectures. We report averages over 10 seeds, with 95% CIs. For visual purposes, the top end of some bars are cropped out.

