# OpenReview forum: "$\text{DT}^\text{2}$: Decision-Targeted Digital Twins"
_ICML.cc/2026/Conference — ICML 2026 regular_

### Official Review · Reviewer_158t · 2026-02-25

**Soundness:** 3
**Presentation:** 3
**Significance:** 3
**Originality:** 3
**Overall Recommendation:** 5
**Confidence:** 3

**Summary:**

This paper proposes an approach that learns ML-based data-driven digital twins (DT) (simulators of a real-world system) that can best support downstream decision-making regarding the ranking of different policies. Instead of fitting one-step transitions as well as possible, a new loss term is proposed that encourages the DT to respect pairwise policy rankings (approximated with FQE in offline settings). Experiments were conducted on a few small synthetic domains, continuous-control benchmarks, and a cancer simulator.

**Compliance With Llm Reviewing Policy:**

Affirmed.

**Final Justification:**

The author rebuttal has resolved all my questions. I encourage the authors to include the new results and discussion in the revised version.

**Key Questions For Authors:**

1. L90-right: do we need to a priori know $\Pi$ and all policies in $\Pi$? Does $\Pi$ need to be a finite set? Is that a realistic assumption?
1. L101-right preference ordering defined by the relation $\prec$ - is this a total ordering based on $J(π)$? how is equality handled?
    - also L352 does Spearman's rank correlation handle cases of ties?
1. L116 "DT-estimated value of π" is defined as an expectation over the DT transition model. In practice, is this estimated by Monte-Carlo rollouts (if so, how many), or otherwise calculated analytically?
1. L155-right Theorem 3.3: need to define "strictly better preference ordering" - if ranking A is strictly better than ranking B, does that mean the set of policy pairs that ranking A is correct is a strict superset of the policy pairs that ranking A is correct? Or simply having more correct pairwise comparisons?
1. L202 Kendall’s correlation and L210 ranking loss: this penalizes each policy pair equally. Have you considered if we might consider low-value policy pairs less important in the loss?
1. L236 Eqn(17) loss definition, and L430 Sec 6.4 effect of $\lambda$, and Fig 5: in these experiments $\lambda$ is never set to 1. Looking at the loss, $\lambda=1$ would mean we forgo simulation fidelity and focus solely on the ranking produced by FQE. What happens to performance when $\lambda=1$?
    - And since we are learning a model to predict FQE ranking, there would be some approximation errors. How does the raw FQE ranking compare to the ranking from DT2 with $\lambda=1$.
1. L384: why the adapted VaGraM DT uses MLP while the rest of the models use RNN/ResNet etc? Does that put this baseline at a disadvantages?
1. L368-right "the average transition MSE" - similar to Fig 3, the model isn't minimizing average MSE, but put its emphasis on where the three policies actions are. Is there a way to quantify this as a weighted MSE, where we'd expect DT2 to achieve lower weighted MSE than NLL baseline?
1. L417 "decision-targeting is not overfitting to the policies observed during training" - is this true in general? or does this require a "good" initial $\Pi$? Is there a way to quantify the "coverage" of the initial $\Pi$
1. L421 Table 2 (and other results): in addition to Regret, consider also reporting top-k regret to show if the ranking can identify the best policy among the top few highest ranked.

**Limitations:**

Yes, limitations are discussed and adequate. No immediate impact is noted.

**Strengths And Weaknesses:**

- Soundness and Presentation: this paper is generally well-written, with clear descriptions of the methods and experiments. There are a few small details that require some clarifications, which I listed under Questions.
- Significance: Learning DTs that can best inform downstream decisions such as comparing policies is an important practical problem, especially for high-stakes domains such as medicine. This paper points out an issue with standard training paradigms that focus solely on general simulation fidelity but overlook the ability to rank policies, and proposes a sensible solution.
- Originality: The proposed method optimizes ranking loss by its differentiable surrogate using the tanh function, which is a good adaptation of existing techniques for training DT models.

---

> ### Author Rebuttal · Authors · 2026-03-31
>
> Thank you for your thoughtful comments and suggestions. We give answers to each of your questions below.
>
> ---
>
> **(A) Knowledge and coverage of $\Pi$ (Q1 & Q9)**
>
> There are many real-world applications where the set of candidate policies is finite and known. For example, in medical settings, treatment options follow standardised clinical guidelines (e.g., specific chemo/radiotherapy schedules). Nevertheless, we do not claim that $DT^2$ will only perform well for ranking policies seen during training.
>
> In fact, we have found that $DT^2$ learns generalised, decision-relevant dynamics. This is shown in $\S 6.3$, and we have now run additional experiments testing performance on more out-of-distribution policies in the $\S 6.2$ environments (see our response to point (D) of `Reviewer oAwR`). From these, we see again that $DT^2$ maintains better ranking performance over standard DTs.
>
> We agree that formalising the "coverage" of $\Pi$ is a relevant direction, which we have noted in $\S 7$ as a direction for future work. It would be interesting to see if we can minimise the policies that need to be seen during training, for efficiency, but also, it would be nice to provide some guidance on what kind of unseen policies we expect to be well-ranked.
>
> **Update:** We will add these further OOD ranking experiments to $\S 6.3$.
>
> ---
>
> **(B) Preference ordering (Q2 & Q4)**
>
> In our experiments, because rewards and state-action spaces are continuous, the probability of two distinct policies yielding the exact same expected return is almost surely zero. Therefore, $\succ$ is a total ordering in this setting. Nevertheless, Spearman's correlation can handle ties, by assigning each tied value the average of the ranks they would have occupied, but we did not need this here.
>
> Also, we define a "strictly better preference ordering" in Theorem 3.3 to mean that the set of correctly ranked policy pairs is a strict superset. This can be seen from our constructive proof, where we find a model where the set of correctly ranked policy pairs is a strict superset of the simulation-optimal model.
>
> **Update:** We will clarify our definition of $\succ$ and Theorem $3.3$.
>
> ---
>
> **(C) DT estimated values (Q3)**
>
> Yes, $J(\pi)$ under the DT is estimated empirically via Monte Carlo rollouts, mirroring how we estimate values in the real environment. In our experiments, we use $500$ rollouts.
>
> **Update:** We will add Monte Carlo details in the Appendix.
>
> ---
>
> **(D) Down-weighting low-value pairs (Q5)**
>
> This is an interesting potential direction. Note that our loss does already have an implicit weighting mechanism, via the $\tanh(\Delta_{ij}/\alpha)$ term, downweighting errors in ranking policies that FQE deems to be very similar. We do not currently further weight this loss based on raw policy values, but this could be done to encourage especially better ranking amongst high-value policies.
>
> ---
>
> **(E) $\lambda$ ablation (Q6)**
>
> When we set $\lambda=1$, we find that training becomes quite unstable, and the ranking ability of the DT does not further increase, while simulation loss grows rapidly. Pure "learning-to-rank" objectives are notoriously difficult to optimise from scratch, especially considering we need to backpropagate through recurrent rollouts to do so. The simulation loss can be seen as a smoothing factor for the ranking-loss landscape, making optimisation stable. We therefore do not recommend large $\lambda$ values, as we say in $\S 6.4$.
>
> In the environments tested in this paper, FQE generally had lower regret than the DT models, and we did not usually see $DT^2$ perfectly replicate the FQE rankings.
>
> **Update:** We will expand $\S 6.4$ to discuss behaviour with $\lambda \approx 1$.
>
> ---
>
> **(F) VaGraM architecture (Q7)**
>
> We used an MLP backbone for the VaGraM model to follow the architectural implementation proposed in the original VaGraM paper. Please also see our response to point (B) of `Reviewer frev`, where we document results for more baselines, again following the architectural choices made in those papers.
>
> ---
>
> **(G) Weighted MSE (Q8)**
>
> This is an interesting insight. While we add the ranking loss, rather than using it to explicitly weight the simulation loss, it does appear in $\S 6.1$ to act the same as a weighted MSE would. We do not believe, however, that there is an easy way to prove equivalence to weighted MSE in general. The intuition, however, that our method automatically discovers the most ranking-relevant weighting, is nice.
>
> ---
>
> **(H) Top-k regret (Q10)**
>
> We will include top-k regret in the final version. For now, you can see the results from Figure 4, but now with top-3 regret, in the figures linked [here](http://anonymous.4open.science/r/DT2-r/pendulum_pipeline_top3.pdf). $DT^2$ outperforms on this metric as well.
>
> **Update:** We will add top-k as an additional ranking metric.
>
> ---
>
> Thank you once again. We hope that we have addressed all your comments, and we would greatly appreciate any further feedback.

---

> > ### Author Rebuttal · Reviewer_158t · 2026-03-31
> >
> > Thank you for the responses to my comments.
> >
> > In point (E) of the response, the authors state that "FQE generally had lower regret than the DT models". I think this is a little concerning, since the overarching problem tackled by the paper is to improve policy ranking and decision regret (which is reflected through the main evaluation metrics). If FQE can already solve this question quite well and outperforms DT-based methods, then it begs the question why DT-based methods are necessary.
> >
> > To address this, I'd encourage the authors to:
> > - Include FQE as a baseline in Fig 4. If FQE indeed outperforms DT-based methods in terms of Spearman correlation and regret, then it should be explicitly acknowledged.
> > - Provide more discussion of why DT-based approaches is still necessary. Is it because it allows for rollouts that can be inspected, whereas FQE just gives a single number without the ability to generate rollouts? Why is this distinction important?

---

> > > ### Author Response · Authors · 2026-04-03
> > >
> > > Thank you for your continued engagement with our work. We appreciate the follow-up question, as it prompted us to formally benchmark FQE and clarify the motivation for our method. We address your comments below.
> > >
> > > ---
> > >
> > > **(A) Empirical comparison with FQE**
> > >
> > > We thank the reviewer for raising this point. Originally, we did not quantitatively compare against FQE because we viewed it as falling under a different decision-making paradigm (see point (B) for reasons). Our statement that "FQE generally had lower regret than the DT models" was made heuristically, based on judgement from early experimentation. However, this should be verified, as we agree that FQE can nevertheless serve as a useful baseline. Prompted by your feedback, we evaluated FQE rankings in our continuous control environments, and have constructed new figures with it included, linked [here](https://anonymous.4open.science/r/DT2-r-r/README.md).
> > >
> > > We do indeed see that FQE performs best, in terms of regret and Spearman's correlation, in two environments  - Hopper and Ant. Interestingly, however, in the remaining four environments one or multiple $DT^2$ configurations actually outperform FQE, but FQE still performs relatively competitively in these. FQE does consistently outperform the traditional simulation-loss DTs. The results are largely favourable for our method, and show that it can outperform FQE in terms of ranking performance. Also, DTs have other benefits that make them particularly useful for human-in-the-loop decision-making processes, which is our concrete focus in this paper, discussed below.
> > >
> > > **Update:** We will include the FQE baseline in $\S 6.2$.
> > >
> > > ---
> > >
> > > **(B) Decision-making with DTs vs. FQE**
> > >
> > > Even in cases where FQE might have a lower regret, DTs possess several distinct, structural advantages that make them more useful for our overarching goal: **human-in-the-loop decision support**.
> > >
> > > FQE would effectively act as an entirely "black box" component when used in a decision-making process, outputting a single scalar value (expected return of $\pi$). While potentially accurate, this offers a human decision-maker no insight into why $\pi$ is good, or how the system is expected to behave, and it is difficult to know when to reject FQE estimates as inaccurate.
> > >
> > > A DT, conversely, outputs a multivariate simulation for each $\pi$ under consideration, uniquely permitting:
> > > 1. Consideration of Subjective/Secondary Criteria: Decisions may involve subjective factors that are not perfectly captured by a formally defined reward function. By generating rollouts of all variables, DTs allow practitioners to assess $\pi$ in terms of some subjective factors, alongside the more formal reward of interest, to enable a comprehensive decision.
> > > 2. Human-in-the-loop "Rejection Sampling": If a user observes a simulation that violates known constraints (e.g., negative blood pressure), they immediately know to trust the model's recommendation less for that specific $\pi$. FQE does not provide this safety check.
> > >
> > >
> > > The above give us motivation in our work to ensure that, in our decision-targeted training method, the simulation fidelity of the DT remains strong, such that the DT can still be used for the above tasks. We do observe that the test-set MSE from $DT^2$ remains relatively good in $\S 6.2$ and $\S 6.3$, and we show that $\lambda$ can effectively be used to trade this off with decision quality ($\S 6.4$).
> > >
> > > Furthermore, there are positives in terms of efficiency in decision-making for DTs over FQE.
> > >
> > > 3. Once trained, a DT can evaluate new, unseen policies, at inference time, just by generating rollouts that they induce. FQE, on the other hand, requires a full training loop to be run for each policy under consideration, which leads to considerably slower decision-making processes, and requires the user to maintain access to the offline dataset. Because DTs are much more efficient in this respect, they can be used more interactively to conduct "what-if" planning, where a user can tweak a policy and quickly observe the simulated outcomes.
> > >
> > > **Update:** We will expand our discussion in $\S 5.2$ on the benefits of DTs over FQE.
> > >
> > > ---
> > >
> > > Thank you once again. We believe these additions will strengthen the paper's narrative and thoroughness. We hope that we have addressed all your comments, and, if so, that you would consider raising your score.

---

### Official Review · Reviewer_frev · 2026-03-03

**Soundness:** 4
**Presentation:** 3
**Significance:** 4
**Originality:** 3
**Overall Recommendation:** 5
**Confidence:** 3

**Summary:**

The authors introduce **DT²**, a Decision-Targeted Digital Twin training paradigm that improves Digital Twin (DT) decision-making by optimising for downstream tasks in addition to typical, context-agnostic machine learning metrics (simulation error, e.g., MSE, NLL, etc.).

Formally, they establish that for any DT that cannot *exactly* model its target system — i.e., some error is guaranteed — models exist that induce better policy rankings than those optimised solely based on simulation error.
The proposed DT² method addresses this by introducing a loss metric catered to preference ordering.
This is enabled 1) by defining a differentiable ranking loss based on an approximation of Kendall’s rank correlation coefficient, 2) by deriving proxy ground truths (real ground truths are intrinsically unavailable) using Fitted Q-Evaluation, a form of off-policy evaluation, and 3) by employing bootstrapping to mitigate computational costs and exploding gradients.

Empirical investigation demonstrate the effectiveness of DT².
First in three toy scenarios, limited hypothetical settings that demonstrate the theoretical validity of the problem statement and the method as the solution.
Then in a more expressive set of benchmark control environments, spanning a range of complexities, in which DT² demonstrates improved ranking alignment and decision regret over naive benchmarks. However, this comes with a modest cost to MSE, which confirms the core hypothesis that optimising for simulation fidelity severs policy ranking.
An additional medical case study mimics these results, but with a negligible MSE cost in this case.
Finally, the effect of the lambda hyper-parameter, responsible for balancing simulation versus decision loss, is tested, suggesting that only a small portion of the overall loss has to be decision-targeted.

**Compliance With Llm Reviewing Policy:**

Affirmed.

**Final Justification:**

An already solid paper that just requires some fine-tuning of the presentation, introduction, and results discussion; the strong rebuttals provide no doubts all this will be addressed.

**Key Questions For Authors:**

1. *"Generating [trajectories for each policy], and determining such policy rankings exactly is generally impossible, as it would require repeated experimentation on the real-world system."*
Could you elaborate on why/when relying on real-world experimentation is infeasible? Or more generally, what’s the motivation behind using Digital Twins as practical surrogates in the first place?  While reasonable to expect readers to know of DTs, the introduction glosses over this ‘why use DTs’ question, which is worth some consideration.

2. Why was the VaGraM method chosen as a benchmark method? More generally, are there other training paradigms to compare the DT² method to?

**Limitations:**

yes

**Strengths And Weaknesses:**

1. Soundness
The paper claims that DTs in decision-making scenarios should be optimised not solely on simulation fidelity, but on a decision-targeted metric as well.
This is very well supported, both through theoretical analysis and empirical results. The supporting theorems appear to be correct and are inherently quite logical. Empirical tests in minimal toy settings clearly verify these theorems. The results from more expressive experiments demonstrate the validity of the approach further, with both positives and negatives shown that confirm the author’s hypotheses.  As only remark, it remains unclear how this novel training paradigm could be applied to or compares to the state-of-the-art.
2. Presentation
The paper is well-structured, with a dense narrative that remains easy enough to follow for informed readers. However, this does not apply to how the work positions itself compared to other literature. The introduction provides only limited context, which is fine as it ought to be expanded on in the related works section, but this is not the case. Section 5 — which for starters should come after the introduction, not after the methodology — is too generic, in particular the subsection on DTs, as it only describes recent works without highlighting gaps to be addressed or differences in approach.
Minor issues:
    - (Figure 1) It gets the point across, but it can be clearer (it took me a few minutes to properly understand). I’d suggest reducing the caption text, have Φ̂1 and Φ̂2 as y-axis labels or subfigure titles, simplify the legend (the two simulations do not need different linestyles), and either add some minimal context to clarify the effects of insulin (on the trajectories), or simplify the labels to reduce cognitive load.
    - (Figure 2) The green colour is too muted, could be clearer. Also, is the “x” next to the Φ outside of F supposed to denote the location of the real-world transition distribution? The use of “x” instead of the white dots for the hypothetical phis is confusing and ought to be equal or more comparable. “x" can also be interpreted as “times Φ” here.
3. Significance
While not completely clear to what extent current DTs are misaligned in for decision-making, the importance of the problem itself is clearly laid out, with various real-world/realistic scenarios described that stand to benefit significantly from DT². The method itself is additionally architecture-agnostic, making it easily applicable in many situations. Moreover, it proves that the prevailing practice of maximising simulation fidelity is suboptimal, informing future works that better models, achievable through DT², exist.
4. Originality
The work appears quite original, combining several techniques to enable tackling the problem that is clearly laid out, advancing the field towards more interpretable DTs to support human decision-making.   While well-justified, it is not completely clear how it compares to closely related literature, DTs in particular.

---

> ### Author Rebuttal · Authors · 2026-03-31
>
> Thank you for your thoughtful comments and suggestions. We give answers to each of your questions below.
>
> ---
>
> **(A) Practical reasons for DTs**
>
> The core motivation for using a DT as a practical surrogate stems from the desire to make optimal decisions (i.e. choosing a policy $\pi \in \Pi$) in situations when the practical risks, costs, and ethical barriers associated with experimentation on the real-world system make trial-and-error infeasible. In fields like medicine, finance, climate policy, or manufacturing, executing policies in the real world can have irreversible consequences, and so there must be some way to deliberate over different options without executing each one.
>
> For example, related to our medical case study ($\S 6.3$), consider how a DT could generally benefit a clinician treating a cancer patient. When choosing a treatment plan for a patient with cancer, it is usually impossible, both ethically and safety-wise, to test multiple different treatments in tandem/one after the other, to see which yields the best outcome, and then choose to continue with the optimal one. Instead, using a DT to simulate the effects of each policy can allow the clinician to rank the efficacy of different treatment options before deployment, and optimise a patient's treatment plan without risking their health with real-world experiments. DTs can serves as practical *in silico* testing grounds to simulate trajectories and establish preference orderings without risking real-world harm. Similar examples also can also be constructed in the other domains, beyond medicine, mentioned earlier.
>
> **Update:** We will **expand the introduction** to clearly **motivate the use of DTs** in real-world settings.
>
> ---
>
> **(B) Added baselines**
>
> **Why VaGraM?** We chose to adapt VaGraM because it is a prominent value-aware Model-Based RL (MBRL) method. As we describe in $\S 5.3$, value-aware MBRL methods are related to our setting, as there share some common themes with our work in terms of motivation and algorithm. Through the comparison with VaGraM (appropriately adapted to our setting), we wanted to test how $\text{DT}^2$ compares to a representative method from this broad field, and also to see whether biasing training of a dynamics model based on some notion of the policy values for $\pi \in \Pi$, through an existing appraoch (VaGraM's value-function-gradient weighting), could achieve comparable benefits as our proposed ranking loss. Our results showed that our method, explicitly targeting pairwise rankings rather than value-function gradient weighting, is far more effective for this specific task.
>
> **Added Baselines:** To provide a more complete comparison to existing works, we have now run additional experiments against a wider suite of baselines. This includes a recent hybrid digital twin baseline (HDTwin (Holt et al., 2024)) and several prominent offline MBRL methods (MOReL (Kidambi et al., 2020), MOPO (Yu et al., 2020), ROMI (Qiao et al., 2026)). As shown in the table linked [here](https://anonymous.4open.science/r/DT2-r/mbrl-and-hdtwin-results.pdf), where we compare the best performing $\text{DT}^2$ and standard DT architectures, per environment, against these new methods, $\text{DT}^2$ clearly outperforms all of these baselines in both decision regret and Spearman's correlation across the majority of the six continuous control environments we consider.
>
> Furthermore, we do think it is important to recognise that the standard DT models (using MSE/NLL losses) already presented in our paper can be seem as a representation of current DT literature, as indeed the vast majority of recent DT works do employ such simulation-based losses (see concrete citations in our introduction). We did demonstrate how many modern architectures (MLPs, Transformers, ResNets, Neural ODEs, RNNs), which are often used as baselines in DT papers (e.g. in (Holt et al., 2024)), can be improved upon using our loss rather than these typical simulation losses.
>
> **Update:** We will **extend our empirical results section** to include these **additional DT and offline MBRL baselines** to better position our method against closely related literature.
>
> ---
>
> **(C) Presentation and Figures**
>
> We appreciate your constructive feedback regarding the presentation. We will make the requested changes:
> * We will refine Figure 1 by simplifying the legend and adding subfigure titles to improve clarity.
> * We will make the colours in Figure 2 more distinct, and replace the "X" notation denoting the position of the true $\Phi$ with a circular marker.
> * We will bring the Related Works section forward to $\S 2$, and expand it to more clearly highlight the gaps in the current DT  literature that our work addresses.
>
> ---
>
> Thank you once again. We hope that we have addressed all your comments, and we would greatly appreciate any further feedback.

---

> > ### Author Rebuttal · Reviewer_frev · 2026-04-03
> >
> > Thank you for the response, nothing more to add 👍

---

> > > ### Author Response · Authors · 2026-04-06
> > >
> > > Thank you for your time and efforts during this review period. We are glad that we have fully addressed your concerns, and we are very grateful for the recommendation for acceptance!

---

### Official Review · Reviewer_YLjT · 2026-03-11

**Soundness:** 3
**Presentation:** 4
**Significance:** 2
**Originality:** 3
**Overall Recommendation:** 4
**Confidence:** 4

**Summary:**

This paper presents a new method to train digital twin in the sequential decision making setting (primarily MDPs). Technically, this is in the same setting as model-based RL but with slightly different purposes. The main idea in the paper is to use the relative ranking of different rollouts as part of the loss function to make sure the digital twin (model) is aligned and useful for the downstream decision-making process. Therefore, the authors propose a relative ranking loss where the ranking is based on the off-policy policy evaluation (OPE, the authors use fitted Q evaluation here for simplicity) to add to the loss function to train the digital twin (model).

To fix the non-differentiability of the ranking operator, the authors relax the strict ranking to soft ranking (sign) by tanh function. To avoid the issue of backpropagating through a long rollout, the authors propose freeze the Q function obtained from the FQE and only do one step Bellman update to obtain a cumulative reward estimate that can backpropagate gradient (Section 4.3). Combining all of these, the authors are able to use a combination of the ranking-based loss and a standard MSE (or any simple simulation loss) to train the digital twin. The authors conduct experiments in one sine function synthetic experiment, mujoco environments, and one cancer treatment experiment. The proposed method can outperform the standard ML training algorithm on the decision regret and relative ranking consistency. The results justify the motivation and claim of the paper.

**Compliance With Llm Reviewing Policy:**

Affirmed.

**Final Justification:**

I am satisfied with the new experiment included in the rebuttal. The new experimental results also support the claims in the paper and are good evidences to strengthen the argument. With that, I have increased the score to 4.

**Key Questions For Authors:**

Nothing specifically, The paper is very well-written and very clear to me. Please respond to the weaknesses mentioned above.

**Limitations:**

Please refer to the weaknesses section.

**Strengths And Weaknesses:**

# Strengths
* The paper is very well-written and clear. It summarizes the literature of several domains well (but with some important related areas missing, which I will mention in the weaknesses part).
* The paper clarifies the motivation very well with illustrative examples.
* The proposed approach (relative ranking in MDP rollouts) is new to me.

------

# Weaknesses:
* The idea is actually not new in MDP and optimization. There is a rich literature of "decision-focused learning" that is trying to address the same issue of misalignment between simulation loss v.s. decision quality/regret. Please find a few references below.
* There are also a few papers extending "decision-focused learning" to the MDP setting, which also tries to handle the same challenge in offline model-based RL (basically the same as digital twin when the problem is offline). It is unclear if the proposed method is better than the literature. But nonetheless, this approach proposed by the authors is still new to me.
* Lack of comparison to model-based RL baselines. Although the authors claim that model-based RL and the digital twin problems are different by their purposes, in fact offline model-based RL (or batch RL) is operating on a fixed set of trajectories collected from one or multiple fixed behavior policy. Several important baselines in offline MBRL are not discussed and not included in the comparison.
* In my opinion, the experimental results make sense, but they convey the same finding/observation as in the literature of decision-focused learning and their MDP extension. This paper does successfully extend to large-scale MDP settings by using the proposed method, but it would still be useful to compare with any valid decision-focused learning approaches and offline MBRL baselines.


If the authors can show that their proposed method can outperform offline MBRL baselines and justify why other decision-focused learning baselines do not apply, I am happy to reassess my evaluation.

-------

# Some references mentioned above

Decision-Focused Learning
* "Task-based End-to-end Model Learning in Stochastic Optimization", Priya Donti, Brandon Amos, J. Zico Kolter, 2017
* "Melding the data-decisions pipeline: Decision-focused learning for combinatorial optimization", B Wilder, B Dilkina, M Tambe, 2019

Decision-Focused Learning in MDPs
* "Popcorn: Partially observed prediction constrained reinforcement learning", J Futoma, MC Hughes, F Doshi-Velez, 2020
* "Learning MDPs from Features: Predict-Then-Optimize for Sequential Decision Problems by Reinforcement Learning", K Wang, S Shah, H Chen, A Perrault, F Doshi-Velez, M Tambe, 2021
* "Decision-Focused Model-based Reinforcement Learning for Reward Transfer", Abhishek Sharma, Sonali Parbhoo, Omer Gottesman, and Finale Doshi-Velez, 2024


Offline MBRL References;
* "MOReL : Model-Based Offline Reinforcement Learning", Rahul Kidambi, Aravind Rajeswaran, Praneeth Netrapalli, Thorsten Joachims, 2020
* "MOPO: Model-based Offline Policy Optimization", Tianhe Yu, Garrett Thomas, Lantao Yu, Stefano Ermon, James Zou, Sergey Levine, Chelsea Finn, Tengyu Ma, 2020

---

> ### Author Rebuttal · Authors · 2026-03-31
>
> Thank you for your thoughtful comments and suggestions. We give answers to each of your questions/concerns below.
>
> ---
>
> **(A) Connection to decision-focused learning**
>
> We thank the reviewer for pointing us toward the literature on decision-focused learning (DFL). We agree that these are relevant works, and we will extend our Related Work section with a detailed discussion of this literature to better contextualise our contribution.
>
> While the foundational non-MDP methods (Donti et al., 2017; Wilder et al., 2019) do not apply to our sequential setting, we will directly address the MDP extensions you mentioned. Firstly, a fundamental distinction between our work and these DFL MDP works is the core objectives. The DFL MDP papers generally target learning a model *such that the optimal policy within it achieves high ground-truth value*. In contrast, our goal, in similar terms, is to construct a model *such that policies from a candidate set are well ranked within it*. Our objective is more relevant to human-in-the-loop decision-making with DTs. Consequently, the training of these DFL methods is quite different to ours.
>
> As well as this shared distinction in problem setting, the specific works cited have technical constraints that make direct empirical comparison inappropriate or infeasible:
> *   **Futoma et al. (2020):** Operates in a POMDP setting, with discrete actions only. Their method is not applicable to the continuous-action environments we consider.
> *   **Wang et al. (2021):** Addresses a fundamentally different problem, attempting to predict MDP parameters (transition or reward parameters) from features of the MDP (e.g. external static descriptions of the MDP), rather than learning a simulator from observed state-action trajectory data.
> *   **Sharma et al. (2024):** Focuses on learning a model that is robust to changes in reward function. They consider access to the ground-truth environment when training, whereas we operate in an offline setting. They also only consider simple linear and tabular MDP settings, leaving scalability to expressive models open.
>
> To our knowledge, $\text{DT}^2$ is the first method to directly optimise an expressive dynamics model for policy ranking amongst a candidate set, bringing the DFL literature to the DT setting. Furthermore, we also focus on maintaining simulation fidelity of the model, for human interpretability purposes, which these DFL MDP works are not generally concerned with—if their models are good in terms of their optimal policy, their objective is satisfied.
>
> **Update:** We will extend $\S5$ to incorporate this discussion of DFL.
>
> ---
>
> **(B) Offline MBRL baselines**
>
> We appreciate the suggestion to compare against offline MBRL. As you noted, offline MBRL also learns a dynamics model from offline trajectory data, and so, although its purpose (using the model to learn an optimal policy) differs from ours (using the model to rank a set of candidate policies), it is possible to compare to some works empirically. Note that many offline MBRL works learn so-called 'pessimistic' dynamics to avoid 'model exploitation', where the optimal policy explores regions that the model cannot accurately predict, and this inductive bias may not be optimal for ranking performance.
>
> We have now run additional experiments comparing $\text{DT}^2$ against prominent offline MBRL methods. Specifically, we compare with the two foundational papers you referenced, **MOReL** (Kidambi et al., 2020) and **MOPO** (Yu et al., 2020), which both train neural ensembles of dynamics models and construct a 'pessimistic MDP', penalising the reward function in transitions where there is high uncertainty. We also compare to a very recent offline MBRL baseline, **ROMI** (Qiao et al., ICLR 2026), which alternates between training an ensemble of neural dynamics models and a SAC policy, and weights the dynamics training using a value-aware approach to incorporate pessimism adversarially, rather than based on uncertainty. Also note that we already included an adapted version of **VaGraM** (Voelcker et al., 2022) in our original submission. Because VaGraM is an *online* MBRL method, we adapted it to be capable offline, and compared with it to see if encoding policy values via value-function gradient weighting (rather than our ranking loss) was beneficial.
>
> The table linked [here](https://anonymous.4open.science/r/DT2-r/mbrl-results.pdf) reports ranking performance of these new baselines and the best performing $\text{DT}^2$ and standard DT architectures, per environment, across the environments from $\S 6.2$. The new baselines are about as competitive as the simulation-loss DTs, and $\text{DT}^2$ outperforms in both regret and Spearman's correlation across almost all environments.
>
>
> **Update:** We will update $\S 6.2$ to include these additional offline MBRL baselines.
>
> ---
>
> Thank you once again. We hope that we have addressed all your comments, and we would greatly appreciate any further feedback.

---

> > ### Author Rebuttal · Reviewer_YLjT · 2026-04-03
> >
> > I thank the authors for the response.
> >
> > To clarify, I value the contribution of using ranking to evaluate the learned model/policy. This is new and a useful objective to me. My major concern is just that I think all the offline model-based RL and the DFL papers should fit into your setting. Regardless whether there is any feature or POMDP v.s. MDP, these methods are all offline model-based setting and they all apply to your setting. The major difference is just that you evaluate the model/policy differently (MSE, pessimistic MSE, OPE, and your ranking-based loss), which I also believe that the proposed method should also easily outperform other offline model-based approach on your metric.
> >
> > I appreciate that you ran a new experiment and the result makes sense to me. It would be great if the authors can include these baselines in other comparisons in the paper (like in Figure 4). The current Figure 4 just shows that DT^2 can lead to improvement on different architectures. A more meaningful experiment would be to show that DT^2 can outperform other methods using different loss functions (like your new table).
> >
> > I am happy to raise my score to 4 if the authors can incorporate the new comparison of different offline model-based RL approaches with some of the different loss functions (MOReL and MOPO -> pessimistic MSE, DFL MDP -> OPE, DT^2 -> ranking loss), which in my opinion is more useful than Figure 4.

---

> > > ### Author Response · Authors · 2026-04-06
> > >
> > > Thank you for your efforts during this review period, and your continued engagement with our work. We are glad that the new experiments are well received, and we will certainly add these new results into the final manuscript. We are also very grateful for the resulting score increase!
> > >
> > > Regarding the Decision-Focused Learning (DFL) baseline:
> > >
> > > While we believe that the specific DFL papers you included cannot be easily applied to the continuous control environments we consider (because they are designed for simple MDPs or discrete action spaces, which can make the ways that they differentiate through the policy optimisation process incompatible), we have identified a more compatible DFL MDP approach, termed *Optimal Model Design (OMD)* ("Control-Oriented Model-Based Reinforcement Learning with Implicit Differentiation", Nikishin et al., 2022). This work uses a method quite similar to the Sharma et al. (2024) paper, but it is more compatible because it is not concerned with generalising to unseen reward functions (which is not our setting), and it applies OMD to more complex environments and transition functions. OMD uses implicit differentiation to optimise the learned model such that the Q-function of the optimal policy within it matches the true Q-function of the optimal policy, and it is demonstrated on MuJoCo environments using deep learning models. We have adapted it to our offline setting, such that we can now compare with a DFL approach. Notably, OMD is an entirely DFL approach, in that it is not concerned with simulation fidelity at all.
> > >
> > > As with the other MBRL baselines, we see that OMD underperforms our $\text{DT}^\text{2}$ method in ranking candidate policies, performing relatively worse than most other methods evaluated. We have now created a Figure-4-style comparison of all of these loss functions across the 6 continuous control environments [here](https://anonymous.4open.science/r/DT2-rebuttals-reply-2-6D34/README.md). This includes comparisons between the following losses:
> > >
> > > 1. Standard NLL
> > > 2. Value-function-gradient weighted MSE (VaGraM)
> > > 3. Pessimistic NLL/MSE via uncertainty (MOReL, MOPO)
> > > 4. Adversarial pessimistic MSE (ROMI)
> > > 5. DFL (OMD)
> > > 6. Our ranking loss ($\text{DT}^\text{2}$)
> > >
> > > In most environments, $\text{DT}^\text{2}$ outperforms in terms of regret (best in 5/6 environments) and Spearman's correlation (best in 5/6 environments). We will add this into the empirical section of our final manuscript, to compare across loss functions, as done here, as well as our initial comparison across architectures.
> > >
> > > Thank you once again for helping us improve our work, these additional results are certainly worthwhile comparisons and they strengthen the paper. Again, we greatly appreciate your time and effort, and are grateful for the resulting increase in score. We hope that we have now fully resolved your concerns.

---

### Official Review · Reviewer_oAwR · 2026-03-12

**Soundness:** 3
**Presentation:** 3
**Significance:** 3
**Originality:** 3
**Overall Recommendation:** 4
**Confidence:** 2

**Summary:**

The paper introduces DT2, a decision-targeted training paradigm for digital twins (DTs). The authors argue that traditional DT training, which minimizes one-step transition errors, can produce suboptimal models for ranking sets of policies, especially when model capacity is limited. They prove this theoretically and show it empirically. To address this, DT2 uses off-policy evaluation (OPE) methods to estimate the values of candidate policies on offline data and encourages the DT to generate rollouts that preserve pairwise policy rankings derived from these proxy ground-truths. The authors demonstrate the efficacy of DT2 across a range of settings and architectures, showing that it consistently improves policy ranking and reduces decision regret relative to conventional DT training, while maintaining a good level of raw simulation fidelity.

**Compliance With Llm Reviewing Policy:**

Affirmed.

**Final Justification:**

Keep original score

**Key Questions For Authors:**

- **Computational Cost**: The authors should provide a more detailed analysis of the computational requirements of DT2. Specifically, they should compare the computational cost of DT2 with conventional DT training and discuss the trade-offs between the two approaches. This would help readers understand the practical implications of using DT2.
- **Interpretability and Transparency**: The authors should provide more information to support their claim that DT2 provides a more interpretable and transparent model compared to OPE methods. Specifically, they could include case studies or user studies to demonstrate the interpretability and transparency of DT2 in real-world decision-making processes.

**Limitations:**

- **Theoretical Assumptions**: The theoretical results (Theorems 3.1 and 3.3) are based on certain assumptions, such as the hypothesis space not containing the true transition distribution (Φ ∉ F) and the existence of local improvements. While these assumptions are reasonable, they may not always hold in real-world scenarios. The authors should discuss the robustness of their results under different assumptions and provide empirical evidence to support their claims.
- **Computational Cost**: The proposed method, DT2, involves additional computational costs, such as the use of off-policy evaluation and the differentiable ranking loss. The authors should provide a more detailed analysis of the computational requirements of DT2 and compare it with conventional DT training. This would help readers understand the trade-offs between the two approaches.
- **Generalization to Unseen Policies**: While the authors show that DT2 can improve policy ranking and reduce decision regret for both seen and unseen policies, they should provide more detailed results on the generalization performance of DT2. Specifically, they should investigate how well DT2 performs in out-of-distribution settings and provide a more comprehensive analysis of its robustness to unseen policies.
- **Interpretability and Transparency**: The authors argue that DT2 provides a more interpretable and transparent model compared to OPE methods. However, they should provide more evidence to support this claim. For example, they could include case studies or user studies to demonstrate the interpretability and transparency of DT2 in real-world decision-making processes.

**Strengths And Weaknesses:**

### 1. Soundness

- **Technical Soundness**: The paper is technically sound. The authors provide a theoretical proof (Theorem 3.1) that standard DT training can produce suboptimal models for decision-making when the hypothesis space is limited. They also provide a second theorem (Theorem 3.3) that shows the existence of better models for decision-making under certain conditions. The empirical results support these theoretical findings, demonstrating the effectiveness of DT2 in various settings.
- **Claims and Support**: The claims are well-supported by both theoretical analysis and empirical results. The authors use a combination of theoretical proofs and empirical evaluations to validate their approach. The empirical results show that DT2 consistently outperforms conventional DT training in terms of policy ranking and decision regret.
- **Methods Used**: The methods used are appropriate. The authors introduce a differentiable ranking loss based on a smooth approximation of Kendall’s rank correlation coefficient, which is a well-established metric for ranking. They also use Fitted Q-Evaluation (FQE) to estimate policy values, which is a robust and widely used method in off-policy evaluation.
- **Theoretical Results**: The proofs provided in the paper are based on reasonable assumptions and are logically sound. The authors clearly state the conditions under which their theorems hold.
- **Empirical Results**: The experiments are well-designed and cover a range of settings, including both limited and expressive hypothesis spaces. The results are consistent and show the advantages of DT2 over conventional DT training.

### 2. Presentation

- **Clarity and Structure**: The paper is clearly written and well-structured. The authors provide a clear introduction to the problem, a formal definition of the decision-making goal, and a detailed explanation of the proposed method. The theoretical and empirical results are presented in a logical and coherent manner.
- **Narrative**: The overall narrative is easy to follow. The authors clearly explain the motivation for their work, the limitations of existing approaches, and the contributions of their proposed method. The paper is well-organized, with each section building on the previous one.
- **Context and Positioning**: The work is properly positioned in the context of prior and concurrent literature. The authors discuss the limitations of existing DT training paradigms and the advantages of their proposed method. They also provide a detailed comparison with related work in the fields of digital twins, off-policy evaluation, and model-based reinforcement learning.

### 3. Significance

- **Problem Addressed**: The paper addresses an important and relevant problem in the field of digital twins. The authors highlight the critical use case of DTs in decision support and show that traditional DT training paradigms can produce suboptimal models for this purpose.
- **Advancement**: The paper advances the understanding and capabilities of digital twins in decision support. The authors introduce a novel training paradigm (DT2) that aligns the generation of simulations with the goal of policy ranking, thereby improving the decision-making capacity of DTs.
- **Impact**: The contributions of the paper have the potential to influence future research and applications in the field of digital twins. The proposed method, DT2, can be applied to a wide range of domains, including finance, climate science, manufacturing, energy, agriculture, robotics, and medicine. The authors show that DT2 consistently improves policy ranking and reduces decision regret, which can have significant practical utility in real-world decision-making processes.
- **Scope of Impact**: The scope of impact is broad, as the proposed method can be applied to various domains and architectures. The authors demonstrate the efficacy of DT2 across a range of settings, showing that it can improve decision-making in both seen and unseen policies.

### 4. Originality

- **New Insights**: The paper provides new insights into the limitations of traditional DT training paradigms and the need for a decision-aware training mechanism. The authors prove that standard DT training can produce suboptimal models for decision-making and introduce a novel training paradigm (DT2) to address this issue.
- **New Methods**: The paper introduces a new training paradigm (DT2) that uses off-policy evaluation methods to estimate policy values and a differentiable ranking loss to encourage the DT to generate rollouts that preserve pairwise policy rankings. The authors also use a smooth approximation of Kendall’s rank correlation coefficient to enable supervision with pairwise comparisons of OPE estimates.
- **Combination of Techniques**: The paper offers a novel combination of existing techniques, including off-policy evaluation, differentiable ranking loss, and bootstrapping. The reasoning behind this combination is well-articulated, and the authors provide a clear explanation of how these techniques work together to improve the decision-making capacity of DTs.
- **Distinction from Related Work**: The contributions of the paper are clearly distinguished from closely related literature. The authors discuss the limitations of existing DT training paradigms and the advantages of their proposed method. They also provide a detailed comparison with related work in the fields of digital twins, off-policy evaluation, and model-based reinforcement learning.

---

> ### Author Rebuttal · Authors · 2026-03-31
>
> Thank you for your thoughtful comments and suggestions. We give answers to each of your questions/concerns below.
>
> ---
>
> **(A) Computational cost**
>
> We agree that analysing the computational trade-offs of $DT^2$ is important. Computational overhead compared to standard DTs is due to the necessary training of the FQE value functions and the $H$-step unrolling to obtain model-estimated policy values.
>
> Concretely, for the continuous control experiments, training $DT^2$ took up to 30 minutes per run, while standard DTs took up to 10 minutes.
>
> However, it is worth noting that $DT^2$ training overhead is comparable to other decision-aware modeling paradigms. For instance, the adapted VaGraM baseline we have compared to also requires the pre-training of value models for each policy $\pi \in \Pi$.
>
> Furthermore, we now compare with further offline model-based RL baselines (please see point (B) of our response to `Reviewer frev` for details). These new models also take longer than the simulation-loss DTs. Specifically, MOReL and MOPO involve training ensembles of dynamics models, and ROMI requires alternating between training a dynamics ensemble and a SAC policy. These models took up to 15, 42, and 90 minutes respectively on the continuous control environments.
>
> **Update:** We will expand upon the computational costs of each method in our discussion section.
>
> ---
>
> **(B) Interpretability**
>
> To concretely compare interpretability between OPE methods and DTs is difficult, because their outputs are very different. Most OPE methods, e.g. FQE, would effectively be an entirely "black box" component when used in a decision-making process, outputting a single scalar value (expected return of $\pi$). While potentially accurate, this offers a human decision-maker no insight into why $\pi$ is good, or how the system is expected to behave, and it is difficult to know when to reject OPE estimates as inaccurate.
>
> A DT, conversely, outputs a multivariate simulation for each $\pi$ under consideration. This can address some of the drawbacks of OPE methods:
> 1. Consideration of Subjective/Secondary Criteria: Decisions may involve subjective factors that are not perfectly captured by a formally defined reward function. By generating rollouts of all variables, DTs allow practitioners to assess $\pi$ in terms of some subjective factors, alongside the more formal reward of interest, to enable a comprehensive decision.
> 2. Human-in-the-loop "Rejection Sampling": If a user observes a simulation that violates known constraints (e.g., negative blood pressure), they immediately know to trust the model's recommendation less for that specific $\pi$. OPE does not provide this safety check.
>
> **Update:** We will expand our discussion in $\S 1$ and $\S 5.2$ on the interpretability benefits of DTs.
>
> ---
>
> **(C) Theoretical assumptions**
>
> We appreciate your careful reading of Theorems $3.1$ and $3.3$. You are correct that the associated assumptions may not always hold in real-world scenarios. However, please note that we have already empirically investigated some settings where these assumptions may be violated, in $\S 6.2$. Here, we used multiple expressive architectures (Transformers, ResNets, Neural ODEs, MLPs, RNNs) as the backbone of DT models, and we saw that $DT^2$ still showed a marked, consistent improvement over standard training. This demonstrates that the core takeaway of our theory—that simulation loss can misallocate capacity for decision-making purposes—remains practically relevant even when using expressive architectures.
>
> ---
>
> **(D) Generalisation to unseen policies $\pi \notin \Pi$**
>
> We agree that out-of-distribution (OOD) generalisation is an important consideration. Please note that we have already shown some evidence for this in $\S6.3$, where we compare evaluate rankings of $11$ unseen policies.
>
> For further evidence, we have now run some additional OOD evaluations on the three smaller environments from $\S6.2$ (Pendulum, LunarLander, Hopper). We compare standard DT and $DT^2$ policy rankings (using the ResNet architecture, for consistency with $\S6.3$) for $5$ OOD policies that take: **Random Action, Constant 0 Action, Constant Minimum Action, Constant Middle Action, and Constant Max Action**. These constant-action policies are less sophisticated than the PPO policies used in $DT^2$ training, and their different visitation distributions and lower returns make them OOD.
>
> From the table linked [here](https://anonymous.4open.science/r/DT2-r/OOD-results.pdf) we see a consistent improvement in ranking performance by $DT^2$. While the difference is, expectedly, more modest than for in-distribution policies, this provides further evidence that $DT^2$ is not overfitting to the training policies, but is learning general, decision-relevant dynamics.
>
> **Update:** We will add these OOD evaluations to $\S 6.3$.
>
> ---
>
> Thank you once again. We hope that we have addressed all your comments, and we would greatly appreciate any further feedback.

---

> > ### Author Rebuttal · Reviewer_oAwR · 2026-04-02
> >
> > The points raised in limitation part are well addressed. Thanks.

---

> > > ### Author Response · Authors · 2026-04-03
> > >
> > > Thank you for the time and effort you put into reviewing our work, your feedback has been very helpful. We are glad that we have fully resolved your concerns, and we would be very grateful if you would consider raising your final score as a result. Many thanks!

---

### Decision · Program_Chairs · 2026-04-30

**Decision:**

Accept (regular)

**Comment:**

The work represents a timely and important contribution to learning digital twins that steers away from traditional transition-focused training losses towards decision-targeted training that preserves pairwise policy rankings.  Experiments clearly identify the advantages of decision-targeted learning across diverse domains.  Post-rebuttal, reviewers unanimously agree on acceptance to ICML.  Reviewers request that the authors incorporate the rebuttal discussion and new results in the final version of their paper.